# Radical Hysterectomy or Total Mesometrial Resection—Two Anatomical Concepts for Surgical Treatment of Cancer of the Uterine Cervix

**DOI:** 10.3390/cancers15215295

**Published:** 2023-11-05

**Authors:** Stoyan Kostov, Pavel Sorokin, Bruno Rezende, Hakan Yalçın, Ilker Selçuk

**Affiliations:** 1Department of Gynecology, Hospital “Saint Anna”, Medical University—“Prof. Dr. Paraskev Stoyanov”, 9002 Varna, Bulgaria; 2Research Institute, Medical University Pleven, 5800 Pleven, Bulgaria; 3Department of Gynecologic Oncology, Moscow City Oncology Hospital No. 62, Istra, 27, Str. 1-30, Moscow Region 143515, Russia; sor-pavel@ya.ru; 4Department of Gynecologic oncology, Londrina Cancer Hospital, Londrina 86015-520, Brazil; bru_rezende_210@hotmail.com; 5Department of Gynecologic Oncology, Ankara Bilkent City Hospital, Maternity Hospital, 06800 Ankara, Turkey; drhyalcin@yahoo.com (H.Y.); ilkerselcukmd@hotmail.com (I.S.)

**Keywords:** nerve-sparing radical hysterectomy, total mesometrial resection, ontogenic anatomy, parametrium, inferior hypogastric plexus, mesometrium

## Abstract

**Simple Summary:**

Cancer of the uterine cervix is the fourth most common cancer and one of the main causes of death in women worldwide. For decades, a radical hysterectomy has been a standard surgical procedure of treatment for women with early-stage cancer of the uterine cervix. However, the disagreement regarding surgical anatomy terms during a radical hysterectomy still raises many controversies among authors, leading to different understandings while performing the procedure. Therefore, Michael Höckel proposed a different and unique surgical concept based on ontogenetic anatomy. The ontogenetic anatomy allows for predicting the potential extent of locoregional cancer spread, which is required to plan adequate surgical treatment. The surgical procedure based on this concept is called total mesometrial resection. The present article aims to describe and compare the anatomical and surgical basics of a radical hysterectomy and total mesometrial resection regarding surgical treatment of cancer of the uterine cervix.

**Abstract:**

A radical hysterectomy is the standard method of surgical treatment for patients with early-stage cancer of the uterine cervix. It was first introduced more than 100 years ago. Since then, various and many different radical procedures, which diverge in terms of radicality, have been described. Inconsistencies are clearly seen in practical anatomy, which were defined as surgically created artifacts. Moreover, the disparity of the procedure is most notable regarding the terminology of pelvic connective tissues and spaces. Despite these controversies, the procedure is widely performed and implemented in the majority of guidelines for the surgical treatment of cancer of the uterine cervix. However, a different and unique concept of surgical treatment of cervical cancer has been reported. It is based on ontogenetic anatomy and maps any tissue in the mature organism according to its embryologic development. The clinical implementation of this theory in the context of early cervical cancer is total mesometrial resection. The present article aims to describe and compare the anatomical and surgical basics of a radical hysterectomy (type C1/C2) and total mesometrial resection. Discrepancies regarding the terminology, resection lines, and surgical planes of both procedures are highlighted in detail. The surgical anatomy of the pelvic autonomic nerves and its surgical dissection is also delineated. This is the first article that compares the discrepancy of classic anatomy and ontogenic anatomy regarding surgical treatment of cancer of the uterine cervix. Clinical data, oncological outcome, and neoadjuvant and adjuvant treatment regarding both procedures are not the topic of the present article.

## 1. Introduction

A radical hysterectomy (RH) is the standard method of surgical treatment for patients with early-stage cancer of the uterine cervix [1]. It was first introduced by Clark at Johns Hopkins Hospital in 1895 [2]. In 1898, Wertheim performed an RH with parametrial resection and described the procedure in detail [3]. Since then, various and many different radical procedures, which diverge in terms of radicality, have been described [4,5,6]. Inconsistencies are clearly seen in practical anatomy, which were defined as surgically created artifacts [6]. Moreover, the disparity of the procedure is most notable regarding the terminology of pelvic connective tissues and spaces. A new classification of RH was introduced by Querleu and Morrow, which tried to define and make uniform the terminology and radicality of the procedure [5]. However, the disagreement regarding surgical anatomy terms during an RH still raises many controversies among authors, leading to different understandings while performing the procedure. Additionally, a different and unique concept with good surgical results has been reported by Höckel [7]. This concept is based on ontogenetic anatomy and maps any tissue in the mature organism according to its embryologic development. The clinical implementation of this theory in the context of early cervical cancer is total mesometrial resection (TMMR) [7]. The aim of the present article is to describe and compare the anatomical and surgical basics of an RH and TMMR. The controversies associated with the clinical data and oncological outcomes of an RH and TMMR are not the topic of the present article. 

## 2. The Anatomy of the Parametrium

The focus of the RH procedure is the resection of the connective tissue attached to the cervix and upper vagina, which is defined as the parametrium. Terms such as “ventral” versus “dorsal”, “cranial” versus “caudal”, and “medial” versus “lateral” will be used in the present article to describe the anatomical positions. However, with regard to the standard anatomical position, the terminology has been appropriately adjusted to improve understanding, such as anterior for ventral and posterior for dorsal. If the surgical supine position (SSP) is used to describe the anatomical structures, it is mentioned [8]. 

The uterine ligaments will be understood through the three-dimensional anatomical model. The term “parametrium”, which is the fatty-lymphoid tissue around the uterus (uterine body and cervix), defines three parametria—the lateral, ventral, and dorsal compartments [8].

The lateral parametrium includes the “parauterine” (uterine artery, superficial uterine vein, and fatty and lymphatic tissue) and “paracervix” tissue (deep uterine vein (vaginal vein), IHP, and pelvic splanchnic nerves), lying between the lateral wall of the cervix and the deep part of the pelvic sidewall. Due to its specific location regarding the ureter, the parauterine tissue and the paracervix will be also termed the “supraureteric parametrium” and “infraureteric paracervix”, respectively [1,5,8]. The parauterine and the paracervix tissues are together called the paracervix by the *Terminologia Anatomica*, which describes the tissue in an en bloc manner [9]. In addition, the cardinal ligament is also the term used to describe the lateral parametrium or paracervix [5,6].

The ventral parametrium is the vesicouterine ligament (VUL), extending between the cervix–upper vagina and bladder, with an anterior (superomedial to the distal ureter in the SSP) and a posterior (inferolateral to the distal ureter in the SSP) leaf, first defined by Okabayashi [4]. The terms vesicouterine and vesicovaginal ligament (VVL) are also used to describe the ventral parametrium for the anterior and posterior leaves of the VUL, respectively [8,10]. On the other hand, neither the term VUL nor VVL is stated in the *Terminologia Anatomica* [9]. However, the pubocervical ligament exists in the *Terminologia Anatomica*, which was introduced to lie between the anterolateral cervix and the deep posterior part of the pubis, and the VUL can be accepted as the proximal part of the pubocervical ligament [9]. The VVL contains the paravaginal veins (it will be also termed the vesical venous plexus). The paravaginal veins or vesical venous plexus have been termed as “middle and inferior vesical veins” and just “inferior vesical veins” and “median and lateral vesical veins” by different authors [11,12,13]. We preferred this terminology, as in the majority of cases it is not just a single or two veins but a venous plexus, which drains into the vaginal vein (deep uterine vein).

The term “paracolpium” will be considered as a separate fatty-lymphoid anatomical structure lateral to the upper vagina and posterior (inferior in the surgical supine position) to the distal ureter. On the other hand, paracolpium is not an official term used in the *Terminologia Anatomica* [9]. However, the paracolpium is referred to as a component of the paracervix [5,9].

The dorsal parametrium includes the uterosacral (cranial) and rectovaginal (caudal) ligaments [5,8,9]. The uterosacral ligament is noticed cranial (anterior) to the ureteric line, and the rectovaginal ligament is noticed caudal (posterior) to the ureteric line [11,12,13,14,15]. The rectovaginal ligament is not stated in the *Terminologia Anatomica*, and the rectouterine ligament is used as a synonym for the uterosacral ligament to define all the dorsal (posterior) parametrial tissue [9].

A conventional RH, according to the Querleu and Morrow classification type C1/C2, and TMMR will be discussed in the present article [5,7]. Additionally, some terms, surgical steps, and borders of resection lines from the Japanese technique for a nerve-sparing RH will also be included [6,11,13]. In accordance with Muallem, we believe that all the parametria could be totally incised with careful dissection and preservation of the nerves [12]. Therefore, we also describe and illustrate the type C2 RH with selective nerve preservation (mainly vesical nerve fibers from the inferior hypogastric plexus)

The anatomy of the three parametria of the uterus is shown in Figure 1 and Figure 2. 

## 3. The Anatomy of the Pelvic Autonomic Nerves (Inferior Hypogastric Plexus)

The superior hypogastric plexus (SHP) is formed by the caudal extension of the nerve fibers from Thoracal (T)10 to Lumbar (L)2/3 and it is located anterior to the left common iliac vein at the cranial part of the retrorectal space. The HN is a bilateral structure that arises from the SHP and carries the sympathetic nerve fibers. The HN runs ventrocaudally, first posterolateral to the rectum at the retrorectal space and then lateral to the rectal sidewall and uterosacral ligament, approximately 2 cm posterior to the ureter at the medial pararectal space, to join the IHP [11,12,13,14].

The pelvic splanchnic nerves (PSNs) originate from the anterior rami of the sacral (S) spinal nerves (S2–S4) and carry the parasympathetic nerve fibers. The PSNs lie posterior to the visceral tributary of the internal iliac vein, the deep uterine vein, at the deep lateral part of the pararectal space. The distal portions of the PSNs run at the caudomedial edge of the ischial spine in the deep paracervix (between the pararectal and paravesical spaces) to join the IHP. The majority of PSNs have a caudomedial course from the pelvic sidewall that crosses the pelvic floor to join the IHP. A few nerve fibers from the PSNs have a retroperitoneal cranial direction toward the mesentery of the sigmoid and descending colon [9,11,12,13,14,15].

The IHP, also known as the pelvic plexus, is located bilaterally at the lateral part of the upper vagina (posterolateral to the upper vagina) and rectum. The IHP can be dissected at the medial part of the paracervix posterior (inferior in the SSP) to the deep uterine vein (vaginal vein). The IHP has a mixed nerve structure by the sympathetic and parasympathetic fibers and is formed by the fusion of the HN, PSNs, and sacral splanchnic nerves. Sacral splanchnic nerves also carry sympathetic fibers to the IHP [11,12,13,14,15].

The bladder (vesical) nerve fibers leave the IHP (IHPvf) in a caudolateral (a few in caudomedial) direction and lie posterior to the distal ureter at the paravaginal area, toward the ureterovesical junction located at the bladder base. The IHPvf run parallel and posterior to the paravaginal veins between the posterior leaf of the VUL and the paracolpium. The IHP has a cross shape formed by the HN, PSNs, and vesical and uterine nerve branches [11,12,13,14,15].

The anatomy of the pelvic autonomic nerves is shown in Figure 3. 

## 4. Avascular Spaces Nearby the Ventral Parametrium during Radical Hysterectomy 

In anatomical terminology, the term “space” describes an area that is surrounded by two independent fasciae and filled with areolar connective tissue. The avascular spaces of the female pelvis can be divided into surgical and anatomical. The existence of fascia, which surrounds the avascular space, is not required in the definition of surgical space [15,16,17,18]. Avascular spaces can be divided into lateral and medial (median). Mainly, there are two lateral spaces—the pararectal and paravesical. These are 3D anatomical spaces. The medial spaces are artificial and located between two visceral structures [16]. They should not be considered 3D spaces. The lateral and medial avascular pelvic spaces were clearly described in our previous report, and there is not much controversy about them [16]. Therefore, only the avascular spaces located near the VP, causing many misunderstandings, will be summarized in detail. 

The VP lies between the vesicovaginal and medial paravesical spaces [11,17,18]. 

The first space, which is encountered near the VP, is the ureteral space. This space is developed after dissection and separation (craniomedial mobilization) of the parauterine tissue (uterine artery/superficial uterine vein) from the ureter. The zone, limited between the uterine cervix–upper vagina and the ureter, posterior to the parauterine tissue, is the key to accessing the ureteral space [17,18]. 

The ureteral space is limited laterally by the medial aspect of the ureter, medially by the anterolateral edge of the vaginal fornix, cranially by the uterine pedicle, and caudally by the VVL (VUL posterior leaf) [17,18,19]. However, the understanding of the ureteral space is inconsistent among surgeons [11,17,18,19]. Zapardiel et al. termed this space the “ureteral tunnel space”, whereas Wang et al. stated that this space should be differentiated from the ureteral tunnel [18,19]. Because the distal ureter is surrounded anteriorly by the VUL and posteriorly by the VVL, the ureteral tunnel is considered as the course of the ureter through the VP and into the bladder [19].

The Yabuki space (Yabuki’s paravaginal space), also called the Fourth space, is delimited laterally by the ureter and medially by the VUL (vesicouterine ligament anterior leaf) [16,17,20]. Yabuki stated that the Fourth space is located between the suspensory and supporting systems. The suspensory system contains the following ligaments: the pubovesical ligament, VUL, rectouterine/vaginal ligament, and rectosacral ligament. The supporting system contains the lamina ligamenti umbilicalis lateralis, the lateral and cardinal ligaments. The supporting system envelopes the suspensory system from the outside [20]. The Yabuki space and Okabayashi’s pararectal space are one entity, as these spaces are in the same horizontal axis, which is divided by the paracervix. They are delimited laterally by the ureter (also the mesoureter and HN for Okabayashi’s pararectal space) and medially by ligaments (VUL for the Yabuki space and RVL for Okabayashi’s pararectal space) [4,16,17]. Querleu et al. termed the Yabuki space the “paravaginal space” and described its boundaries [15]. However, the Fourth (Yabuki) space and Okabayashi’s paravaginal space are different surgical spaces located near the VP.

Okabayashi’s paravaginal space is encountered after a transection of the VUL. This space divides the upper vagina and paracervical connective tissue into two components: laterally, the VVL (or the VUL posterior leaf), and medially, the lateral vaginal wall (or paracolpium) (Figure 4) [13,17]. Okabayashi’s paravaginal space is bounded cranially by the VUL [13,17].

Our comment on the boundaries of Yabuki’s and Okabayashi’s paravaginal spaces is as follows:

Yabuki’s and Okabayashi’s paravaginal spaces define a similar zone, which is developed after a lateral dissection of the distal ureter from the anterolateral part of the upper vagina. Yabuki’s space is cranial to Okabayashi’s paravaginal space. The Yabuki space is developed at the caudal edge of the parauterine tissue between the VUL and the ureter. Dissection of the Fourth space provides lateralization of the ureter and safety transection of the VUL. Yabuki’s space gives limited access to the VVL (vesicouterine ligament posterior leaf) compared to Okabayashi’s paravaginal space. Okabayashi’s paravaginal space is developed after lateral dissection of the most caudal part of the distal ureter from the vagina, which provides access to the VVL and paracolpium [13,16,17,20].

The development of Okabayashi’s paravaginal space provides total lateralization of the ureter (to the level of the pelvic sidewall) and enables separation of the VVL and paravaginal veins from the paracolpium and IHPvf. By the way, with a meticulous dissection, IHPvf can be preserved (type C1), or the paracervix with the paracolpium structure can be excised totally (type C2) [13]. Between Yabuki’s and Okabayashi’s paravaginal spaces, there is a fibrous tissue attaching the ureter to the lateral vaginal wall, which may contain small vaginal veins [13,16,17,20].

Fuji’s space is another entity that is delimited medially by the upper vaginal wall, laterally by the bladder nerve branch of IHP, cranially by the cut end of the vaginal vein (deep uterine vein) and uterine nerve branch of the IHP, and caudally by the IHP. Fuji’s space is dissected in order to separate the IHP from the lateral vaginal wall and to transect the uterine nerve branch from the IHP [11,12]. Some of the avascular spaces of the pelvis are shown in Figure 4.

## 5. Discrepancy Regarding Terminology, Anatomical Description, and Surgical Technique during Radical Hysterectomy (Type C1/C2)

### 5.1. Dorsal Parametrium

The dorsal parametrium becomes visible after a surgical dissection of the rectovaginal and medial pararectal spaces. The dorsal parametrium contains two parts, the uterosacral (cranial part) and the rectovaginal ligament (caudal part), which lie on the same axis. Grossly, the dorsal parametrium extends from the posterolateral part of the cervix and upper vagina (pericervical adventitia) to the pelvic parietal fascia over the sacral bone and tendinous arch of the levator ani [5,8].

Disagreements arise between authors about the exact terminology and anatomical insertion points of the cranial part of the dorsal parametrium. Some authors define the cranial part as the “uterosacral ligament”, whereas others call it the “recto-uterine ligament” or “dorsal parametrium” [1,5,8,10,15]. Moreover, Querleu et al. went further and stated that the uterosacral ligament does not exist as a ligamentous structure. The authors discussed that the dorsal connection of the cervix does not attach to the sacrum or any bone structure, and the only connective tissue of the female genital tract, which reaches the sacral plane, is the autonomic nerve plexus [15]. Correspondingly, two studies reported that the uterosacral ligament directly attaches to the lateral side of the rectal fascia or presacral fascia, respectively, without connection with the sacral bone [21,22].

The uterosacral ligament joins the sacrospinous ligament/coccygeus muscle and the piriformis muscle, the sciatic foramen, and less commonly the ischial spine [23]. Most data reported that the uterosacral ligament attaches from the dorsolateral portion of the intersection point of the uterine cervix and upper vagina to the presacral fascia at the S2-S4 vertebra level [24,25]. The uterosacral ligament is located lateral to the rectum and medial to the ureter and HN [5,25]. It consists of a superficial fibrous component near the cervix and a peritoneal areolar connective component near the rectum [25]. Therefore, it is preferable for the cranial part of the dorsal parametrium to be termed the “uterosacral ligament” or “rectouterine ligament”.

The caudal part of the dorsal parametrium is defined as the rectovaginal ligament, also called the “sacrovaginal ligament”, “rectal pillar”, or “deep uterosacral ligament” [1,5,25]. Most authors define it as the “rectovaginal ligament” [9,26]. The rectovaginal ligament extends from the dorsolateral portion of the upper and middle vagina to the caudolateral portions of the upper rectum and attaches to the extensions of the tendinous arch of the levator ani and pelvic parietal fascia (pelvic floor fascia). It is thicker and stronger when compared to the uterosacral ligament [10,23,26]. On the other hand, the uterosacral (rectouterine) ligament is commonly used to define all the dorsal (posterior) parametrial tissue [9].

Ceccaroni et al. added the lateral ligament of the rectum as a part of the dorsal parametrium [24]. Actually, the ligament has no connection with the uterine cervix, as it extends from the caudal portion of the internal iliac vessels (ischial spine and proximal part of the tendinous arch of the levator ani) to the mesorectum [24,26]. The lateral ligament of the rectum and the middle rectal artery are useful landmarks during a nerve-sparing RH, as they run closely to the PSNs [24]. However, our surgical experience indicates that the lateral ligament of the rectum is not a component of the lateral or dorsal parametrium to be resected during an RH. Unlike the middle rectal artery (frequency ranges from 20 to 30%), the lateral ligament of the rectum is a constant anatomical structure at the caudomedial pararectal space [26,27].

The mesoureter is a mesentery-like structure formed by the fatty neurovascular tissue, which is located posterior to the ureter and contains the HN [5,8,26,28]. The term ureterohypogastric fascia is used for the tissue plane containing the ureter and the hypogastric nerve as the mesoureter [6,11,18]. After the initial dissection of the posterior leaf of the broad ligament, the mesoureter can easily be detached from the rectouterine and rectovaginal ligaments by entering Okabayashi’s (medial) pararectal space [4,11]. The HN can be dissected lateral to the uterosacral (rectouterine) ligament, and the HN together with the dorsocaudal part of the IHP can be dissected lateral to the rectovaginal ligament [4,11].

We consider the resection margin of the dorsal parametrium during radical hysterectomy (type C1/C2).

#### 5.1.1. Transverse 

Type C1/C2—Dorsal transection of the uterosacral and rectovaginal ligament at the level of the rectum and pelvic parietal fascia covering the upper part of the sacrum [5,8].

#### 5.1.2. Longitudinal (Caudal)

Type C1—The HN and the mesoureter are dissected from the uterosacral (rectouterine) ligament and caudally from the rectovaginal ligament and retracted laterally. At the caudal border, the IHP is dissected and preserved while excising the rectovaginal ligament. The caudal limit of resection of the dorsal parametrium depends on the resection plane of the vagina. Generally, the uterosacral ligament is totally resected, whereas the rectovaginal ligament can be excised partially according to the vaginal resection line. Nevertheless, the rectouterine and rectovaginal ligaments are excised [5,8,11,12,13].

Type C2—Caudal resection of the dorsal parametrium is at the level of the pelvic parietal fascia, which is caudal to the rectal attachments. The HN, part of the PSNs, and the IHP are sacrificed [5,8].

### 5.2. Lateral Parametrium

The lateral parametrium (LP) becomes visible after the dissection of the paravesical (medial paravesical space) and pararectal spaces [8]. The LP is a perivascular sheath that extends from the lateral aspect of the cervix (pericervical adventitia) to the dorsolateral part of the pelvic sidewall (the ischial spine or the proximal part of the tendinous arch of the levator ani) or roughly to the origin of the internal iliac vessels [25]. 

The LP still raises misconceptions and confusion in oncogynecological surgery. The LP has been termed by different names throughout the years—“cardinal ligament” (CL), “Mackenrodt’s ligament”, “transverse cervical ligament”, “retinaculum uteri”, “lateral paracolpium”, or “the web” [1,24,25]. The anatomists do not study the LP in detail, as the development of the pararectal and paravesical spaces is necessary for its clear identification [29]. Two studies discussed that the cranial part of the LP could not be recognized as a ligamentous connective structure, whereas another study reported that the whole LP is not a ligament as it contains condensed connective tissue only medially and fatty-lymphatic tissue laterally [15,19,29]. Conversely, an anatomical study reported that the caudal (inferior in the surgical supine position) part of the LP could be termed as a “ligament” as it is composed of fatty tissue, vessels, nerves, and lymph nodes [21]. In a recent study, Yabuki stated that two lateral parametria exist, the cardinal ligament and transverse cervical ligament, which should be distinguished [30].

Based on the majority of the medical studies in the literature, one might suggest the following separation of the LP:

The cranial part of the LP, which is located anterior to (superior in the SSP) the ureter between the uterine corpus and cranial part of the pelvic sidewall, is called “parauterine”. The uterine artery/superficial uterine vein and the related connective, fatty, and lymphatic tissues are the elements of the parauterine tissue [10,15,31].

The caudal part of the LP, which is located posterior to (inferior in the SSP) the ureter and connects the uterine cervix–upper vagina with the caudal dorsolateral part of the pelvic sidewall (ischial spine or proximal part of tendinous arch of levator ani), is called the “paracervix” [5,10,15,24]. The ureter and the deep uterine vein (vaginal vein) divide the paracervix into subcompartments. The ureter separates the paracervix into medial and lateral parts. The medial part represents a condensation of the connective tissue (pericervical adventitia), whereas the lateral part contains mainly lymph nodes [5,15]. The lymphatic tissue lateral to the longitudinal axis of the internal iliac vessels, posterior to the obturator nerve, and anterior to the gluteal vessels with the sacral nerve plane are also known as “paracervical lymph nodes”, which are removed during a type B2 or type C1/C2 RH. They should be distinguished from the pelvic lymph nodes [5,15]. 

The deep uterine vein is a term that is not officially recognized by *Terminologia Anatomica* [9]. Furthermore, this vein is not described in anatomical textbooks. The term “deep uterine vein” is often encountered in medical articles describing RH [4,6,11,13]. The so-called “deep uterine vein” is actually the vaginal vein [1,6,8,17]. Our surgical experience shows that the vaginal vein is the visceral tributary of the internal iliac vein, which drains the vesical veins and paravaginal veins into the internal iliac vein. The deep uterine vein (vaginal vein) is located approximately 1–2 cm posterior to (below) the uterine artery and superficial uterine vein [8]. In addition, the deep uterine vein (vaginal vein) also lies posterior to (inferior in the SSP) the ureter. The deep uterine vein (vaginal vein) divides the paracervix into the vascular (cranial—above the deep uterine vein) and nervous (caudal—below the deep uterine vein) parts. The vascular part consists of the inferior vesical and vaginal arteries, while the nervous part includes the distal branches of the PSNs and the IHP at the caudomedial part of the pararectal space [5,17,19].

We consider the resection margin of the lateral parametrium during radical hysterectomy (type C1/C2).

#### 5.2.1. Transverse 

Type C1/C1—The lateral resection line is at the level of the internal iliac artery [5,8,10,11,12,13].

#### 5.2.2. Longitudinal (Caudal)

Type C1—The deep (caudal) resection margin is at the level of the deep uterine vein (vaginal vein) in order to preserve the underlying PSNs and IHP [5,8,10,11,12,13].

Type C2—This is a complete resection of the LP down to the level of the pelvic floor fascia (medial aspect of the gluteal and pudendal vessels, corresponds to the level of the ischial spine and proximal part of tendinous arch of the levator ani). The paravesical and pararectal spaces become one entity, and the PSNs and IHP are sacrificed [5,8,11,12,13].

### 5.3. Ventral Parametrium 

The ventral parametrium (VP) becomes visible after dissection of the medial paravesical space and the vesicouterine/vesicovaginal spaces. The vesicovaginal space is dissected to the level of the trigone of the urinary bladder [17]. The VP extends from the anterolateral aspect of the cervix and upper vagina to the bladder at the ureterovesical junction. The VP is divided by the course of the ureter into the anteriorly (superomedial in the SSP) located VUL (vesicouterine ligament anterior leaf) and posteriorly (inferolateral in the SSP) located VVL (vesicouterine ligament posterior leaf) [1,5,8,15]. 

A precise understanding and description of the surgical anatomy of the VP was unclear for more than a century. However, Fujii et al. described in detail the anatomy of the ventral parametrium in 2007 [11,17]. Nevertheless, surgical misconceptions still exist regarding the terminology and anatomy of the VP, especially its caudal part. Different terms, such as “bladder pillar”, “anterior mesoureter”, “lateral ligament of the bladder”, “ventral parametrium”, or “ventral paracolpium”, have been used [1,5,8,24]. 

The VUL is the cranial and medial portion of the VP; it lies anterior to the distal ureter, and the term “vesicouterine ligament- anterior leaf” is used as a specific definition. The VUL can be visualized after the dissection or craniomedial separation of the uterine artery/superficial uterine vein from the ureter. The VUL is a constant connective tissue layer extending from the anterolateral part of the uterine cervical fascia to the urinary bladder. It covers the ureter anteriorly and contributes to the anterior portion of the ureteral tunnel (roof of the ureteral tunnel) [17,18,32]. The VUL contains mainly vessels—cervicovesical vessels, the superior vesical vein, and the ureteral and vaginal branches of the uterine artery (anatomical variations may exist where the vaginal artery originates from the uterine artery). The ureteral branch of the uterine artery should be separated and ligated just before the dissection of VUL [1,17]. Our surgical experience in RH has shown that, to resect the VUL from the level of the urinary bladder, the dissection of the VUL should follow the 1 o’clock position anterior to the ureter on the right side and the 11 o’clock position anterior to the ureter on the left side. Thus, the ureter is left at the medial side for developing a safe cleavage of paravaginal space between the anterolateral vaginal wall and the ureter.

The VVL becomes visible after the separation and division of the VUL. The term “vesicouterine ligament- posterior leaf” is also used for the VVL. Okabayashi’s paravaginal space is developed so that after lateral dissection of the ureter, the VVL that is lying at the caudal part of the medial paravesical space posterior to the ureter is fully exposed [4,13]. The VVL is a connective tissue layer that extends from the anterolateral part of the upper vagina to the urinary bladder caudal to the ureterovesical junction. It consists of paravaginal veins (vesical venous plexus) and fatty-lymphatic tissue overlying the bladder nerve branches of the IHP [13,17,18,28]. The anatomical architecture of the VVL is debatable. Some data stated that the VVL contains a fascial component, which originates from the merging of urogenital hypogastric fascia and umbilicovesical fascia [28]. However, other authors concluded that the VVL is not an actual ligamentous structure, as it is mainly composed of vessels and nerves [21,32]. 

The term “paracolpium” has many different anatomical descriptions and concepts in the medical literature [1,5,6,8]. *Gray’s Anatomy* defines paracolpium as a continuation of the parametrium down along the vagina [33]. Some authors emphasized that the paracolpium and the VVL are different entities, with the paracolpium as the fatty tissue lateral to the upper and middle vagina at the caudal level of the VVL, whereas others considered the VVL and the fatty-lymphoid tissue lateral to the upper vagina as one anatomical structure called the “paracolpium” and/or the paracolpium as a part of the paracervix [5,10]. Furthermore, another study divided the paracolpium into ventral, dorsal, and lateral parts [1]. Some Japanese and European authors define the paracolpium as an independent anatomical structure that is integrated with the lateral wall of the upper vagina [6,17,18]. The paracolpium is limited craniolaterally by the VVL [17,18]. There is a vein in the paracolpium, which is parallel to the vaginal wall and medial to the paravaginal veins [17,18]. We termed it the “paracolpium vein”.

Our surgical experience indicates that the paracolpium should be evaluated as a distinct entity from the paracervix. The paracolpium is located caudal to the paracervix and lateral to the upper-middle vaginal wall. The VVL (vesicouterine ligament posterior leaf) and the paracolpium can be dissected and named together or separately. Our surgical experience suggests differing the VVL and paracolpium. The VVL is located caudal to the ureterovesical junction between the anterolateral part of the upper vagina and bladder. On the other hand, the paracolpium lies caudal to the VVL, and the paracolpium can be noticed after lateral dissection of the VVL. The IHPvf lie between the paracolpium and the VVL [13].

We consider the resection margin of the ventral parametrium during a radical hysterectomy (type C1/C2).

#### 5.3.1. Transverse

Type C1—This is a partial dissection of the ureter from the VUL and anterolateral vaginal wall. The VUL is transected medial to the ureter. By the way, the VUL (anterior leaf) can be excised close to the level of the urinary bladder (approximately 1-2cm distal to the vaginal wall) [5,8,11,12,13].

Type C2—This is a complete resection of the VUL (anterior leaf) from the level of the urinary bladder. It is a complete lateral dissection of the distal ureter from the anterolateral vaginal wall. By the way, the VVL and paracolpium can be fully excised after developing the paravaginal space [5,8,11].

#### 5.3.2. Longitudinal (Caudal)

Type C1—The VVL (vesicouterine ligament posterior leaf) is excised. The limit of resection is the IHPvf, which are preserved. After developing Okabayashi’s paravaginal space, a meticulous dissection is required to separate the bladder branches of the IHP between the paracolpium and the VVL. The paravaginal veins (vesical venous plexus) are identified, cut, and ligated [8,11,12,13].

Type C2—The resection line depends on the level of the vaginal cuff resection. The paracolpium (fatty tissue lateral to the vaginal wall) is resected together with the vaginal tissue. The IHPvf are not identified, and the VVL with the paracolpium is excised en bloc [5,8,11,12,13]. 

Some of the surgical steps of a type C1 RH and type C2 RH with selective nerve preservation are shown in Figure 5 and Figure 6.

## 6. Introduction to Ontogenetic Anatomy

Ontogenetic anatomy is a relatively novel concept based on embryologic development. During early embryologic development, the proliferating groups of cells do not mix with each other. They form units—ontogenetic compartments—which can be identified and mapped in mature organisms [34,35]. The ontogenetic cancer field model was developed by Höckel and considered malignant tumor growth as an anisotropic process. According to the theory, tumor growth is confined by a permissive compartment during the long time of its natural history, and the compartment’s borders are tumor suppressive. It is also a predictive model, which contributes to selecting the proper surgical procedure for every single patient according to the probability of the compartment’s involvement. [35,36]. 

The term “meso” is widely used in ontogenetic surgery. It is an abbreviation of “mesentery”, which differs from the classical anatomical definition. In general, the term “mesentery” means the structure attaches the organ to the posterior abdominal wall and is covered by the peritoneum on both sides. In ontogenetic anatomy, the prefix “meso-” defines a compartment derived from distinct primordial tissue. Mesotissues abut the corresponding organ, although their origin differs from the organ [35,36]. It contains fibro-fatty tissue, blood vessels, lymphatics, and nerves. Sometimes, meso-structures are enveloped by an easily dissectible fascia (e.g., mesorectum and mesorectal fascia); in other cases, it could not be dissected in an avascular plane and the related veins may communicate between different compartments (e.g., the Mullerian compartment and mesobladder) [35,36]. 

### 6.1. Ontogenetic Anatomy and Total Mesometrial Resection

Any tissue in a mature organism can be identified according to its developmental mapping. In the female pelvis, the embryological margins and tissue planes of these compartments are more or less apparent [34,35,36]. One of the most obvious borders is the mesorectal fascia. The mesorectal fascia envelopes the mesorectum circumferentially and can be sharply dissected. The mesorectal ontogenetic compartment is formed from the hindgut—the posterior part of the alimentary canal (Figure 7) [37]. The clinical translation of this knowledge is the total mesorectal excision (TME)—the current standard of rectal carcinoma treatment [38]. 

Some discrepancies exist between conventional and ontogenetic anatomy concerning TME. The lateral rectal ligament and middle rectal artery have been accepted by some surgeons as a part of the rectal anatomy with no doubt [19,39]. However, when TME becomes the main procedure in rectal cancer surgery, these structures are no longer mentioned during mesorectal dissection. According to development, the hindgut is located intraperitoneally, and there is no middle rectal artery as a branch of the internal iliac artery, and there is no lateral ligament of the rectum [39,40]. 

The urogenital sinus (part of the ventral cloaca) gives rise to the urinary bladder, urethra, and distal 1/3 vagina. The intrapelvic part of this compartment has a well-demarcated border. Paramesonephric (Mullerian) ducts are differentiated into the fallopian tubes, uterus, cervix, and proximal 2/3 of the vagina. The most complex part of the Mullerian compartment is located subperitoneally. Some parts of the subperitoneal compartment are covered by an enveloped fascia, which is well demarcated, but some other parts are dissected from other compartments by their proper fascia [35,36,41] (Figure 8). 

Two pairs of wings—vascular and ligamentous mesometria—are noticed. These mesotissues (especially vascular mesometria) contain intercalated lymph nodes and should be removed during a TMMR. Despite the proximity to the Mullerian compartment, mesometria originate from other compartments: vascular mesometria from anterior cloacal mesenchyme and ligamentous mesometria from distal splanchnic coelom. In the context of TMMR, a few parts of the urogenital sinus derivatives should be sharply dissected from the Mullerian and mesotissues compartments: the dorsal bladder adventitia from the cervix and the bladder mesentery from the vascular mesometrium. The vascular mesometrium contains the uterine artery and fibro-fatty and lymphatic tissues. This structure is similar to the LP during a type C RH. Ligamentous mesometrium is comparable with dorsal parametrium, which includes the most dorsal part close to the sacrum and also the deep or caudal part, called the rectovaginal ligament. The ligamentous mesometrium can be easily dissected from the edge of the mesorectal fascia [35,36,41]. 

The next part of the subperitoneal Mullerian compartment is the proximal 2/3 of the vagina and mesocolpium. The distal part of the vagina is formed from the urogenital sinus and is no longer a part of the TMMR procedure [7]. However, if a clear vaginal margin can be achieved without extirpation of the whole Mullerian vagina, an intracompartmental transection is appropriate to preserve the vaginal function [35,36].

The ureter is located below (caudal to) the vascular mesometrium and laterally to the ligamentous mesometrium. It develops as an outgrowth from the mesonephric duct and opens into the bladder (part of the urogenital sinus). According to developmental anatomy, the ureter and its mesentery—the mesoureter—can be sharply dissected from parts of the Mullerian compartment [42].

The HN is located between the mesoureter and the ligamentous mesometria proximally [35,36,41]. This concept differs from the Japanese one, where the mesoureter contains the HN [6,11,13]. Distally, the hypogastric nerve fuses with the PSNs and they together form the IHP. The IHP is located between the mesoureter and mesocolpium. For distal mesoureter mobilization, vaginal branches of the IHP should be transected and the bladder branches are preserved [35,36,41]. Conventional anatomy describes mesoureter as a lamina close to the rectum and without a clear ventral border at the ureterovesical junction [8,30]. In ontogenetic anatomy, the mesoureter can be dissected at the ventral side of the vascular mesometrium and up to the bladder. Höckel preserves the hypogastric nerve plane during total mesometrial resection [35,36,41].

The onion model was suggested by Höckel to explain ontogenetic anatomy (Figure 9). In contrast to conventional anatomy, this model is devoid of dissection artifacts—spaces and ligaments. According to the ontogenetic anatomy, the only ligaments that exist are derived from the gubernaculum—round ligaments and utero-ovarian ligaments [35,36,41]. 

### 6.2. Differences between the Ontogenetic Anatomy and Traditional Anatomy for Radical Hysterectomy

The ontogenetic anatomy has some discrepancies when compared to the traditional anatomy. Surgical anatomy for RH is based on the principle of uterocentricity. Any tissues located close to the cervix are called the parametrium or paracervix. Some independent data suggest the absence of the infraureteral parametrium (paracervix), which is part of a type C RH [43]. According to the ontogenetic approach, the LP consists of three different parts: the vascular mesometrium, bladder mesentery, and mesoureter (Figure 10). The dorsal parametrium can focus on ontogenetically different tissues—the mesoureter, hypogastric nerve, partial mesorectum, and ligamentous mesometrium. The VP, except its most proximal part, does not exist according to ontogenetic anatomy. It is a part of the bladder adventitia, and it belongs to the urogenital sinus compartment and does not contain intercalated lymph nodes related to the uterine cervix [35,36,41]. Data from the conventional RH anatomy are also contradictive, and some of these do not confirm lymphatic flow through the VP [44,45]. As a result, in the ontogenetic anatomy and TMMR, our surgical experience shows that the dissection of the VUL should be performed at the 11 o’clock position anterior to the ureter on the right side and at the 1 o’clock position anterior to the ureter on the left side to get a safe margin of a mesoureter dissection from the anterolateral vaginal wall. By the way, during a TMMR, the distal portion of the VUL is not excised, and the VVL with the paracolpium located posterior (caudal) to the distal ureter at the paravaginal zone are also not excised [35,36,41]. An intraoperative view during a TMMR is shown in Figure 11.

Some surgical landmarks and steps during a TMMR are shown in Figure 12.

The differences regarding the terminology between RH and TMMR are shown in Table 1.

## 7. Discussion

### 7.1. Differences in the Surgical Applications of Radical Hysterectomy and Total Mesometrial Resection

Enhanced knowledge of the anatomy is essential during every gynecological surgical procedure. The complexity of the anatomical system of the female pelvis requires meticulous dissection, isolation, and resection. Consequently, the surgical maneuvers and division of the organs are possible by developing the natural avascular areas [15,16,26]. They are artificial and do not exist without surgical intervention [15]. The majority of surgeons concluded that the dissection of these pelvic spaces is an integral part of most pelvic surgical procedures [6,15,18,24]. Additionally, a precise dissection of pelvic avascular spaces will delineate the anatomical boundaries during an RH. Furthermore, the development of these avascular planes decreases bleeding during an RH. The conventional RH has tremendously developed in recent years. The precise anatomy of the VP, a new classification system, and a detailed explanation of a nerve-sparing RH have been described [5,11,12]. These new data provide additional knowledge of the surgical anatomy during an RH. The rising interest in the nerve-sparing RH will definitely improve the quality of life of patients. However, the misconception still exists regarding anatomical terms and surgical techniques during an RH [6]. Consequently, a conventional RH is a surgical procedure based on surgical anatomy rather than theoretical concepts. Despite its acceptance worldwide, almost every surgeon understands and performs an RH in a different manner. Modern classification by Querlue and Marrow categorized RH into more definitive landmarks [5,8]. However, due to the complex 3D anatomy of the parametrium, it is not quite possible to identify precise landmarks, particularly for the VP, and the surgical practice may differ between surgeons. The resection margin of the VVL can be easily misunderstood, particularly for the type C1 RH.

TMMR does not consider the existence of uterine ligaments (except round and ovarian ligaments) and pelvic spaces. Höckel stated that the ontogenetic anatomy allows for describing pelvic structures without dissection biases, and this approach may simplify difficult subperitoneal pelvic anatomy rather than creating dissection artifacts [35,36,41]. However, unlike surgical approaches for rectal and colon cancer, TMMR is not accepted worldwide. The main obstacle is the absence of prospective multicenter trials with long-term follow-up. However, recently published data from the multicenter TMMR register reproduced good oncological results [46]. Moreover, Höckel et al. recently published new data on approximately 500 patients treated with TMMR and a follow-up of five years. The disease-free survival for 5 years was 89.4% [47]. 

Höckel proposed that the distal (sacral) part of the ligamentous mesometria (dorsal parametrium) contains intercalated lymph nodes that drain to the pelvic sidewall (presciatic region and near gluteal vessels), which can be one of the common sites of recurrence [35,36,47,48]. Both types C1 and C2 RH excise the dorsal parametrium (uterosacral and rectovaginal ligaments) totally, and the tissue plane is separated from the mesorectum. For a type C1 RH, the HN and IHP, which are lateral to the rectouterine and rectovaginal ligament, respectively, are dissected together with the mesoureter and preserved [5,8,11,12,13]. Höckel also preserves the HN plane and the IHP during mesoureter mobilization while performing TMMR [35,36,47].

During a type C2 RH, after total lateralization of the ureter to the level of the urinary bladder, all the paracervix and paracolpium tissues are excised. This ventral resection in the type C2 RH also resects the distal mesoureter in which the IHPvf are sacrificed [5,6,8]. In TMMR, the distal ureter lamella plane (mesoureter) is preserved, and the mesoureter preservation results in a decrease in morbidity compared to a type C2 RH [35,36,47,49]. During the ventral resection step of TMMR, the ligation of the vaginal vein and vesicovaginal anastomoses is made medial to the ureter, exposing the resection plane to transect only the uterine nerve branches and keeping the bladder branches of the IHP enveloped with the distal mesoureter. However, the controversy in this issue is the resection of the deep lymphatic tissues draining the upper vagina and lower part of the cervix. The resection of the VVL and paracolpium provides a dissection of the vesicovaginal vessels and corresponding lymphatic channels. The VVL is excised in both type C1 and C2 hysterectomies and during a type C1 hysterectomy, and the IHPvf are selectively preserved [5,8,11,12,13]. According to the new update of the Querleu and Morrow classification of RH, only the medial part of the VP is transected [50]. Cibula mentioned that only 1-2 cm of the VP could be transected in the transverse direction [8]. According to some Japanese and European authors, the whole VP could be transected during a type C1 RH after identification of the IHPvf [8,11,12,13,50].

The parauterine tissues (cranial part of the LP) along the uterine artery are excised during a TMMR, whereas the VVL and the lateral part of the paracervix (transection only medial to the ureter in the transverse direction) are not transected during a TMMR [35,36,41,47]. The paracervix along the deep uterine vein (vaginal vein) contains lymph nodes, besides the lymphatic and venous drainage of the VVL (VUL deep part) tissues toward the deep uterine vein. These are important routes of lymphatic spread in cervical cancer. The paracervix is totally removed during a C2 RH and partially removed during a type C1 RH (to the level of the deep uterine vein). In conclusion, the longitudinal excision of the paracervix is similar in a C1 RH and TMMR—at the level of the deep uterine vein. However, in the transverse direction, the paracervix is transected more laterally during a C1 RH (at the level of hypogastric vessels) compared to a TMMR (medial to the ureter) [5,8,35,36,41,47].

An RH and a TMMR show worse patient outcomes (higher recurrence rate and inferior overall survival) when they are performed by a minimally invasive approach compared to open abdominal surgery [46,51]. Höckel always performs TMMR by open surgery [35,36]. Moreover, one study reported that minimally invasive TMMR (laparoscopic and robotic) is associated with a higher recurrence rate compared to open surgery [46]. The Laparoscopic Approach to Cervical Cancer Trial (LACC trial) showed that a minimally invasive RH was associated with inferior disease-free survival and overall survival compared to an open abdominal RH for patients with cancer of the uterine cervix (stages IA1 with lymphovascular invasion, IA2, IB1, and histologic subtypes of squamous-cell carcinoma, adenocarcinoma, or adenosquamous carcinoma) [51]. However, the debate about the role of a minimally invasive RH for the treatment of cervical cancer is still ongoing [52,53]. Nevertheless, an open RH is the standard approach in most centers [54].

Postoperative specimens after a type C1 RH and TMMR are shown in Figure 13.

Injury to the autonomic pelvic nerves during an RH and lymph node dissection is shown in Table 2.

The differences in the parametrial resection between an RH (type C1/C2) and TMMR are shown in Table 3.

### 7.2. Lymphatic Drainage of the Cervix and Upper Vagina with Surgical Implications

The cervix and endometrium have two major lymphatic drainage patterns: firstly, the upper paracervical or preureteral pathway, by means of the uterine artery to the external iliac and obturator nodes; secondly, the lower paracervical or retroureteral pathway, by means of the uterine vein to the hypogastric and presacral nodes. The prospective sentinel lymph node study by Geppert et al. revealed the lymphatic drainage pathway through the upper rim of the uterosacral ligament to the presacral area, medial to the internal iliac vein [55].

The study by Bonneau et al. evaluated the lymphatic and nerve structure of the parametrium [56]. They defined the tissue anterior (superior in the surgical supine position) to the ureter along the uterine artery, the LP. The tissue extends from the lateral side of the cervix and vagina to the lateral side of the rectum, posterior (inferior in the surgical supine position) to the ureter, the posterior parametrium. The paracervix and paracolpium, as well as the uterosacral ligament, are the elements of the posterior parametrium, lying oblique dorsally and caudally. The upper cephalic part of this posterior parametrium is fused with the uterosacral ligament and is attached to the uterus–cervix junction and the rectum, defined as the proximal posterior parametrium. The caudal part of this posterior parametrium attaches to the sacrorectal plane and forms the distal posterior parametrium. Undertaking an immunofluorescence evaluation, Bonneau et al. found that the LP and proximal posterior parametrium consist of lymphatic tissue predominantly; on the other hand, the distal posterior parametrium has significantly less lymphatic tissue. The autonomic nerve structure was found to be higher in the proximal posterior parametrium, lateral to the rectum and vagina [56].

Ercoli et al. defined the lymphatic channels anterior to the ureter along the uterine artery, the supraureteral paracervical pathway. The lymphatic channels along the deep uterine vein, the infraureteral paracervical pathway, are similar to the proximal posterior parametrium in the study by Bonneau et al. [44,56]. The infraureteral paracervical lymphatic channels drain into the interiliac and gluteal/pudendal nodes. In addition, they defined the neural paracervical pathway, which is between the deep uterine vein cranially and the pelvic floor caudally and contains the hypogastric nerve plexus. The cadaveric dissection study by Ercoli et al. revealed that the lymphatic channels of the uterine cervix are mainly found at the supraureteral paracervical pathway (96% of cases), less at the infraureteral paracervical pathway (22% of cases), and rarely at the neural paracervical pathway. Ercoli et al. could not document the lymphatic drainage pathway at the VUL and distal uterosacral ligament. This study evaluates the lymphatic drainage channels by cadaveric dissection only, not examining the tissue histology; by the way, this will be a restriction in findings [44].

The study by Kraima et al. evaluated the parametrial lymphatic structure of female fetuses by immunohistochemistry [45]. During embryological growth, the horseshoe-shaped fascia around the uterine cervix provides a border zone between the bladder and cervix that keeps the ureter and the vesical nerve bundles of the IHP at the lateral side. Similarly, the mesorectal fascia provides a clear border between the rectum and the vagina. Kraima et al. found the supraureteral lymphatic pathway along the uterine artery as one of the major lymphatic pathways. Secondly, a dorsal pathway was found medial to the IHP at the proximal part of the uterosacral ligament. The drainage of the upper vagina was primarily in the dorsolateral angle toward the uterosacral ligament. Kraima et al. did not find lymphatic drainage of the uterine cervix at the deep VUL (VVL) [45].

Benedetti-Panici et al. microscopically analyzed the surgical specimens of early and locally advanced cervical cancer patients to identify the metastatic and non-metastatic lymph nodes in the anterior, lateral, and dorsal parametria [57,58]. They found lymphatic metastatic nodes predominantly in the LP along the uterine artery, as well as in the superficial and deep layer of the VUL (VVL), and in less quantity at the uterosacral ligament. One hypothesis for the results of this study is that the gross cervical tumor changes the direction of the lymphatic flow, and newly formed lymphatic channels will be found in the parametrial tissue, especially near the upper vagina at the deep portion of the VUL (VVL).

The data of the prospective Senticol I and II studies on sentinel lymph node biopsy for cervical cancer were analyzed by Balaya et al. [59]. They found the upper paracervical pathway as the dominant lymphatic drainage of the uterine cervix. The main finding of this study was that a cervical tumor ≥20 mm has an impact on the drainage pathway and significantly reveals atypical drainage patterns. Atypical lymphatic drainage pathways were found in 24.5% of patients with early-stage cervical cancer, and 8.9% of positive sentinel lymph nodes were in the atypical area. This patient group may have atypical lymph channels and nodes in the parametrial tissue apart from the supraureteral pathway [59].

It should be also stressed that during an RH, pelvic and paraaortic lymphadenectomy is a staging procedure and can be replaced by a sentinel lymph node biopsy [54]. Moreover, the current guidelines for the management of patients with cervical cancer recommend abandoning the hysterectomy in case of lymph node metastases [54]. Contrarily, Höckel stated that lymph node dissection during TMMR is therapeutic [35,36,47]. The intraoperative frozen section of lymph nodes is mandatory according to the TMMR algorithm, and if metastases are found, the next line of lymph nodes must be removed. After a lymphadenectomy, the TMMR is performed irrespective of lymph node status [47].

These data showed that more studies of the lymphatic drainage of the uterine cervix are needed in order to estimate which of the procedures (RH or TMMR) provides a better oncological outcome.

However, in order to perform cervical cancer surgery, the lymphatic drainage pathways from the uterine cervix should be carefully analyzed. The embryogenesis should be kept in mind for tissue planes and lymphatic metastasis; however, the cervical tumor size may affect the lymphatic pathways and form new lymph drainage channels at the upper vagina and paracervix.

## 8. Conclusions

A conventional RH and TMMR are two different surgical procedures based on classic surgical anatomy and ontogenic anatomy, respectively. Both procedures raise many questions in terms of anatomical terminology, resection lines, radicality, and tumor disseminations. A convectional RH is more popular worldwide. Therefore, it is implemented in the majority of guidelines for cervical cancer surgery. However, the procedure still provokes misconceptions regarding the terminology about spaces and uterus ligaments. Thus, the discrepancy confuses surgeons concerning the exact resection line during surgery. Controversially, the TMMR describes the terminology and radicality based on ontogenic anatomy. Therefore, differences regarding the terms are less apparent. However, the procedure is less popular and rarely found in cervical cancer guidelines, although it shows good oncological results. If further studies showed a similar oncological outcome, the implementation of the procedure in cervical cancer guidelines could be discussed.

## Figures and Tables

**Figure 1 cancers-15-05295-f001:**
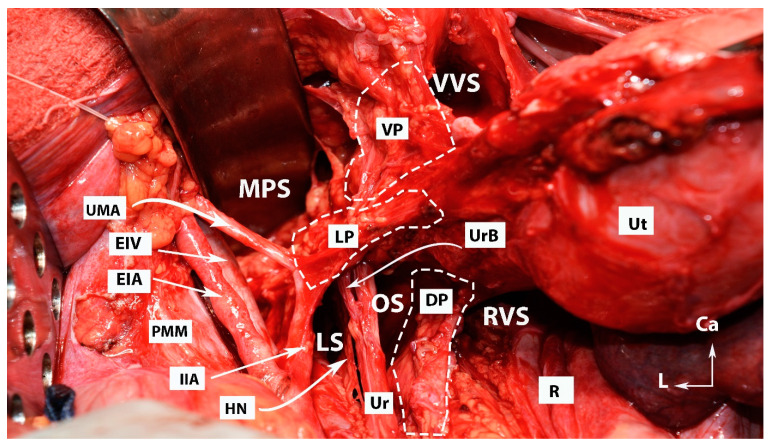
All three parametria after dissection of some of the female avascular spaces in the pelvis (open surgery; author’s own material). VVS—vesicovaginal space; MPS—medial paravesical space; RVS—rectovaginal space; OS—Okabayashi’s pararectal space; LS—Latzko’s pararectal space; VP—ventral parametrium; LP—lateral parametrium; DS—dorsal parametrium; Ut—uterus; R—rectum; UMA—umbilical artery; EIV—external iliac vein; EIA—external iliac artery; PMM—psoas major muscle; IIA—internal iliac artery; HN—hypogastric nerve; Ur—ureter; UrB—ureteral branch of uterine artery; Ca—caudal; L—left.

**Figure 2 cancers-15-05295-f002:**
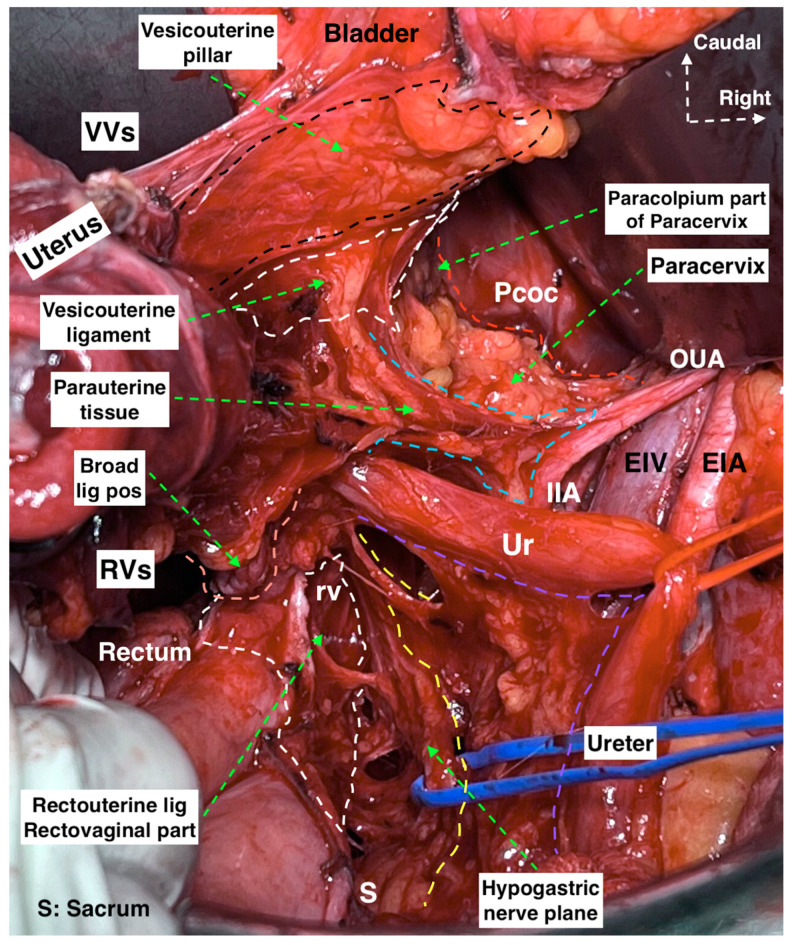
The concept of the lateral (parauterine-paracervix), dorsal (rectouterine-rectovaginal), and ventral (vesicouterine-paracolpium) parametrium during radical hysterectomy with the hypogastric nerve plane. (Surgical dissection by author IS.) (EIA: external iliac artery, EIV: external iliac vein, IIA: internal iliac artery, Ur: ureter, OUA: obliterated umbilical artery, Pcoc: pubococcygeus muscle, VVs: vesicovaginal space, RVs: rectovaginal space, rv: rectovaginal, lig: ligament, pos: posterior, S: sacrum).

**Figure 3 cancers-15-05295-f003:**
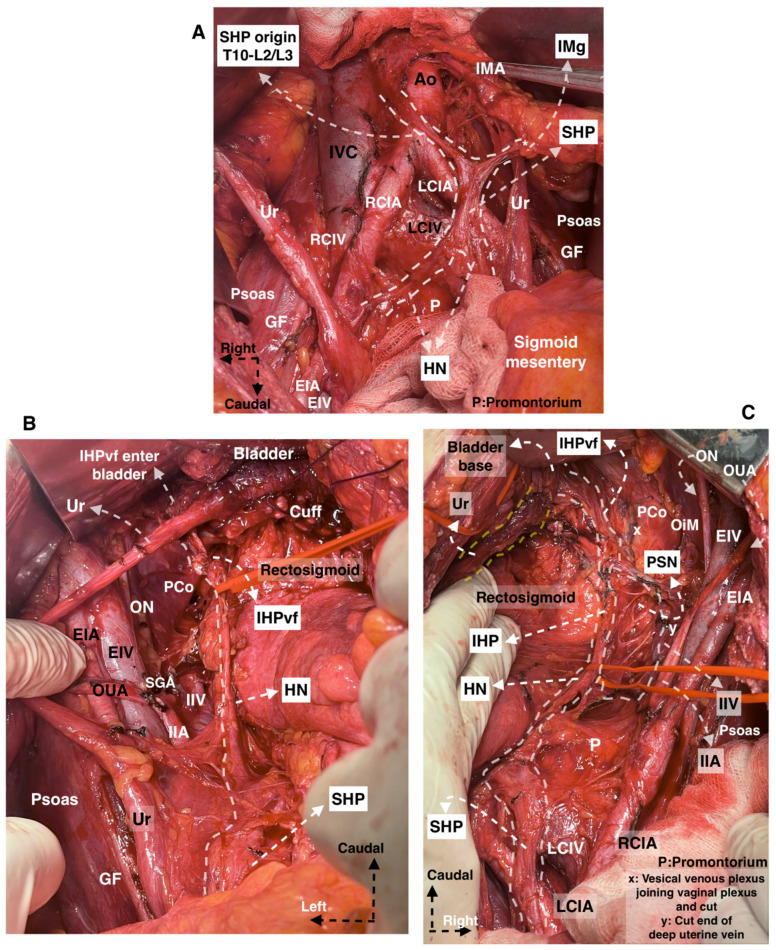
Pelvic autonomic nerve plexus, selective nerve sparing during type C2 radical hysterectomy. The patient had spontaneous voiding on the postoperative 7th day and had no early and late complications. (Surgical dissection by author IS.) (**A**) The nerve fibers originating from Thoracal (T)10 to Lumbar (L)2 with the contribution of nerve fibers from L3 forming the superior hypogastric plexus. The autonomic nerve fibers extend between the SHP and inferior mesenteric ganglion, caudal to the inferior mesenteric artery. (**B**,**C**) The pelvic nerve plane called the hypogastric nerve plane on the left and right side, respectively. Following the nerve line of the hypogastric nerve will lead to the identification and dissection of the pelvic splanchnic nerves and inferior hypogastric plexus vesical fibers (IHPvf). (Ao: aorta, IVC: inferior vena cava, IMA: inferior mesenteric artery, RCIA: right common iliac artery, LCIA: left common iliac artery, RCIV: right common iliac vein, LCIV: left common iliac vein, Ur: ureter, GF: genitofemoral, EIA: external iliac artery, EIV: external iliac vein, SHP: superior hypogastric plexus, HN: hypogastric nerve, IMg: inferior mesenteric ganglion, P: promontorium, IIA: internal iliac artery, IIV: internal iliac vein, SGA: superior gluteal artery, OUA: obliterated umbilical artery, PCo: pubococcygeus muscle, IHPvf: inferior hypogastric plexus vesical fibers, ON: obturator nerve, OiM: obturator internus muscle, PSN: pelvic splanchnic nerves, IHP: inferior hypogastric plexus).

**Figure 4 cancers-15-05295-f004:**
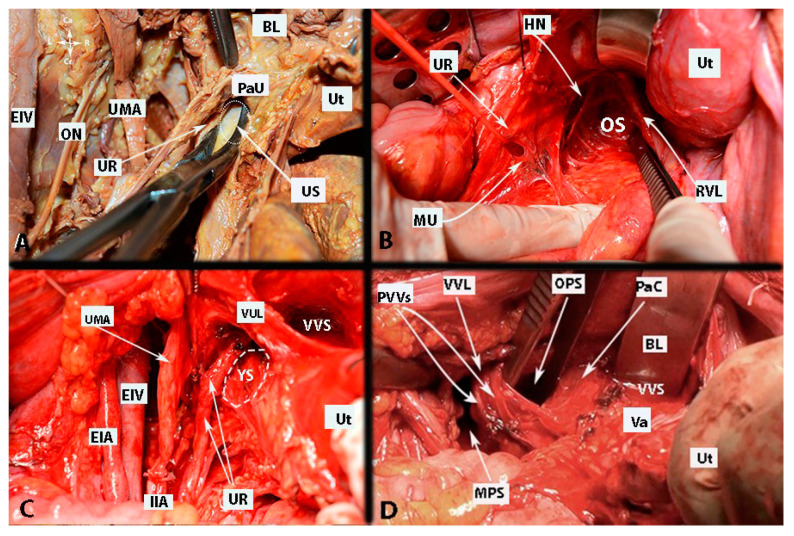
Avascular spaces of the pelvis. Left side of the pelvis—author’s own material. (**A**)—ureteral space; (**B**)—Okabayashi’s pararectal space; (**C**)—Yabuki’s space; (**D**)—Okabayashi’s paravaginal space. EIV—external iliac artery; EIV—external iliac vein; IIA—internal iliac artery; UR—ureter; Ut—uterus; PaU—parauterine tissue (supraureteric parametrium); US—ureteral space; Va—vagina; ON—obturator nerve; BL—bladder; UMA—umbilical artery; VVL—vesicovaginal ligament; VUL—vesicouterine ligament; RVL—rectovaginal ligament; OS—Okabayashi’s pararectal space; YS— Yabuki’s space; OPS—Okabayashi’s paravaginal space; PVVs—paravaginal veins; PaC—paracolpium; VVS—vesicovaginal space; MPS—medial paravesical space.

**Figure 5 cancers-15-05295-f005:**
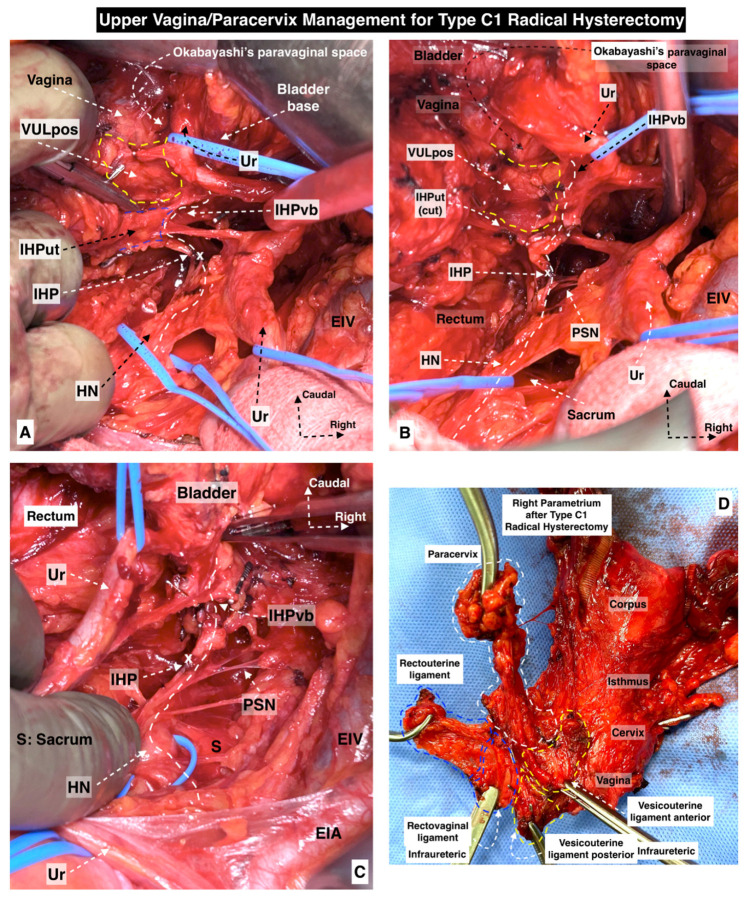
Upper vagina/paracervix management for type C1 radical hysterectomy ((**A**–**C**) show the dissection order and surgical steps, (**D)**—postoperative specimen). The paravaginal space is developed, so the infraureteric paracervix, the posterior leaf of the vesicouterine ligament, is resected superior to the inferior hypogastric plexus vesical branches. (Surgical dissection by author IS) (Ur: ureter, HN: hypogastric nerve, IHP: inferior hypogastric plexus, IHPut: inferior hypogastric plexus uterine branches, VULpos: vesicouterine ligament posterior leaf, IHPvb: inferior hypogastric plexus vesical branches, EIV: external iliac vein, PSN: pelvic splanchnic nerves, EIA: external iliac artery, S: sacrum).

**Figure 6 cancers-15-05295-f006:**
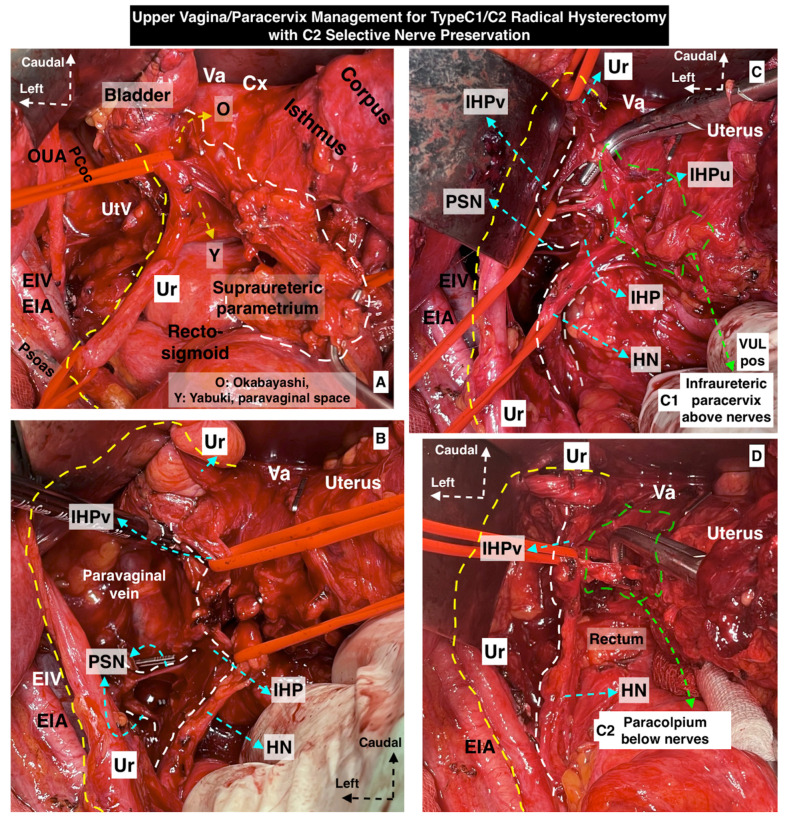
Upper vagina/paracervix management for type C1/C2 radical hysterectomy with C2 selective nerve preservation (**A**–**D** show the dissection order and surgical steps). After identification of the inferior hypogastric plexus vesical branches/fibers by developing the Okabayashi and Yabuki paravaginal spaces, the vesicouterine ligament posterior leaf above the nerves for type C1 radical hysterectomy and paracolpium below the nerves for type C2 radical hysterectomy can easily be resected. (Surgical dissection by author IS) (O: Okabayashi, Y: Yabuki, EIA: external iliac artery, EIV: external iliac vein, OUA: obliterated umbilical artery, Pcoc: pubococcygeus muscle, UtV: uterine vein, Ur: ureter, Va: vagina, Cx: cervix, PSN: pelvic splanchnic nerves, HN: hypogastric nerve, IHP: inferior hypogastric plexus, IHPv: inferior hypogastric plexus vesical branches, IHPu: inferior hypogastric plexus uterine branches, VULpos: vesicouterine ligament posterior leaf).

**Figure 7 cancers-15-05295-f007:**
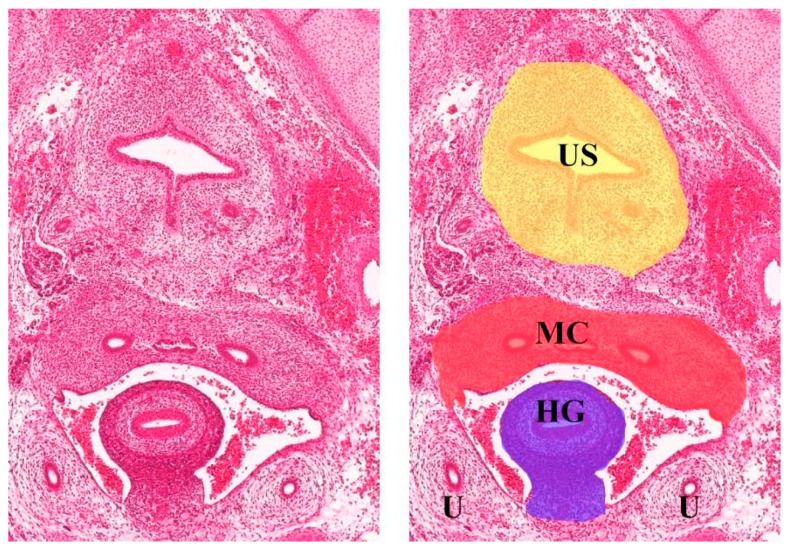
Transverse section of Carnegie stage 23 female embryo (the human histology images were provided by the Joint MRC/Wellcome Trust (MR/R006237/1) Human Developmental Biology Resource (www.hdbr.org, accessed on 8 June 2023; Gerrelli et al. (2015)) reference [37]. HG—hindgut, MC—Mullerian compartment, US—urogenital sinus, U—ureter.

**Figure 8 cancers-15-05295-f008:**
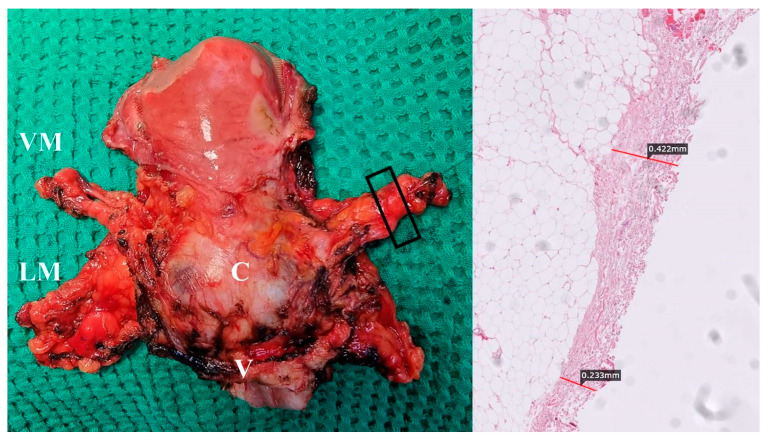
Total mesometrial resection specimen (author’s own material). VM—vascular mesometrium, LM—ligamentous mesometrium, C—cervix, V—vagina. Right picture (black box on left figure) shows fascia that partially covers vascular mesometria.

**Figure 9 cancers-15-05295-f009:**
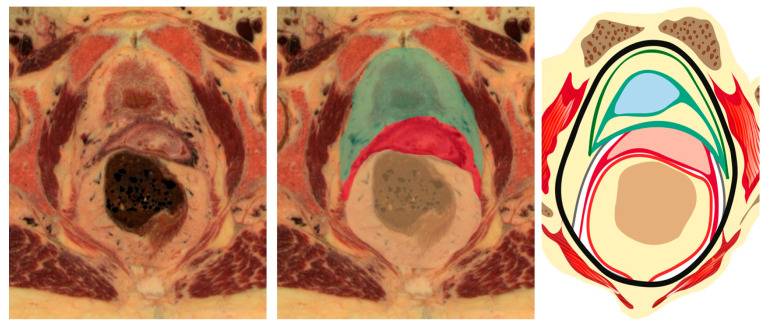
Onion model of female pelvis (author’s own material). Green—urogenital sinus derivate, red—mullerian compartment, yellow—hindgut derivate, gray—ureter and mesoureter.

**Figure 10 cancers-15-05295-f010:**
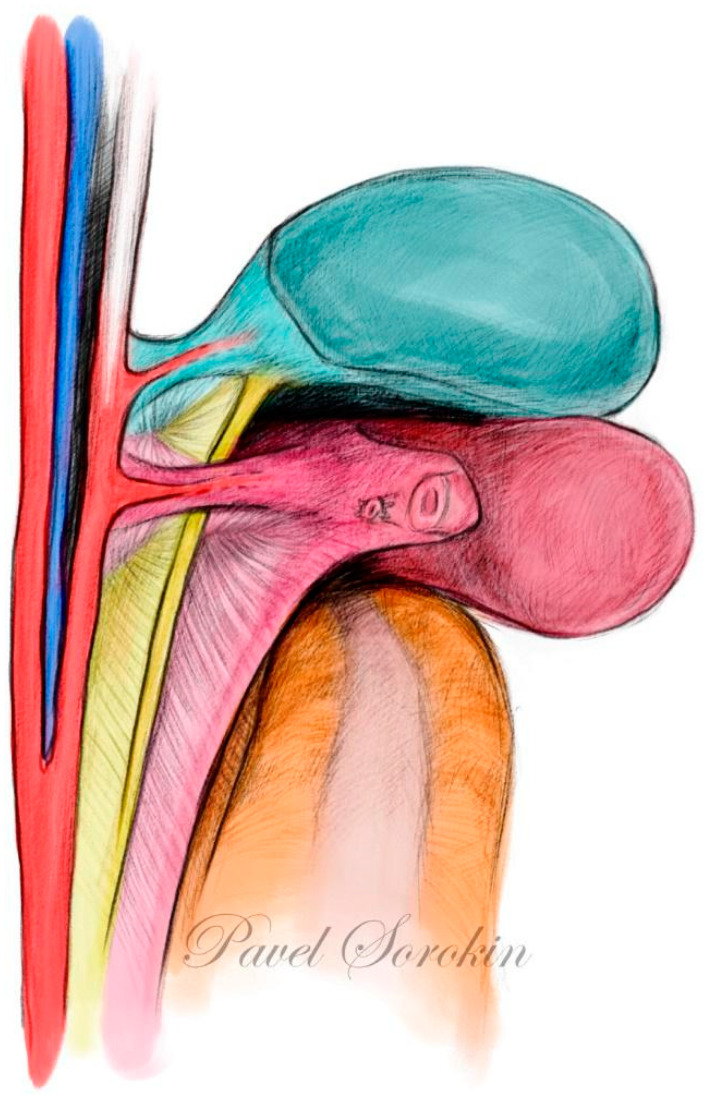
Ontogenetic anatomy for TMMR. Three-dimensional pelvic anatomy model (author’s own material). Lateral parametrium (left side—author’s own material) is split into 3 parts: vascular mesometrium (red), mesoureter (yellow), and bladder mesentery (green). Ventral parametrium is not shown as it is dissection artefact according to ontogenetic onion model.

**Figure 11 cancers-15-05295-f011:**
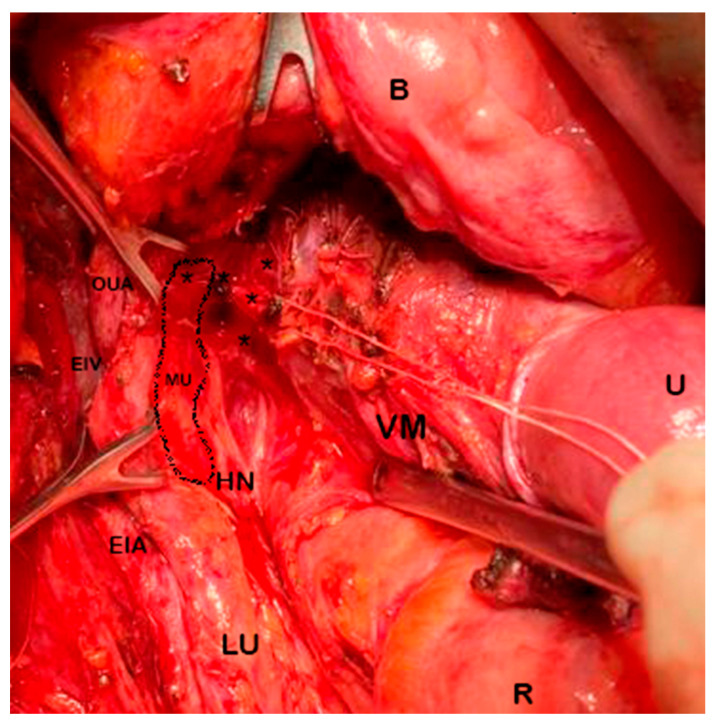
Total mesometrial resection—intraoperative view (left pelvic sidewall—author’s own material). U: uterus; B: bladder; R: rectum; LU: left ureter; EIA: external iliac artery; EIV: external iliac vein; HN: hypogastric nerve; VM: vascular mesometrium; OUA: obliterated umbilical artery; MU: mesoureter demarcated; (*): uterine branches from inferior hypogastric plexus repaired with cotton. Vascular mesometrium is grasped by forceps and guided dorsally. Vaginal veins are ligated and transected medial to mesoureter. Distal mesoureter, hypogastric nerves, and inferior hypogastric plexus are preserved.

**Figure 12 cancers-15-05295-f012:**
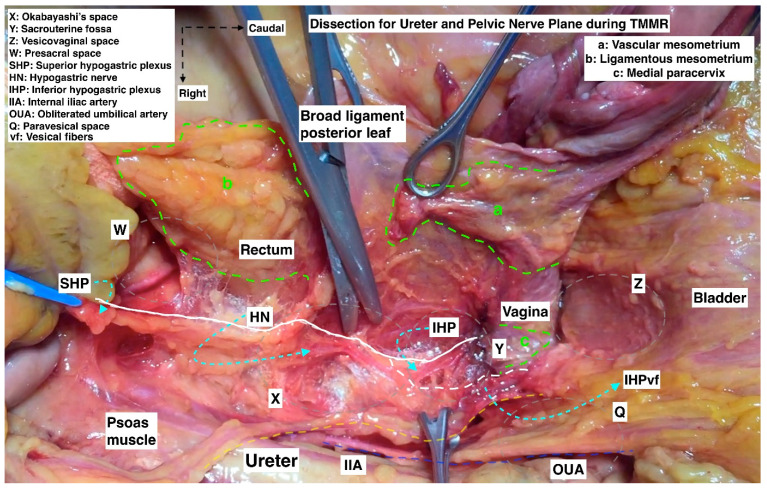
Dissection for ureter and pelvic nerve plane/plexus during total mesometrial resection (TMMR) (cadaveric dissection by Michael Höckel and author IS) (SHP: superior hypogastric plexus, HN: hypogastric nerve, IHP: inferior hypogastric plexus, IHPvf: inferior hypogastric plexus vesical fibers, IIA: internal iliac artery, OUA: obliterated umbilical artery, a: vascular mesometrium, b: ligamentous mesometrium, c: medial paracervix).

**Figure 13 cancers-15-05295-f013:**
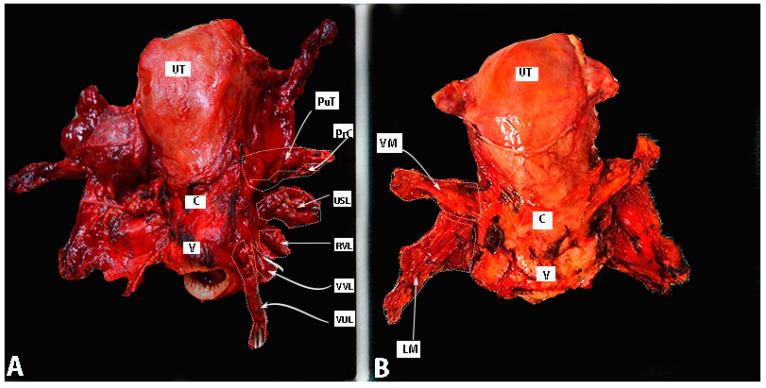
Postoperative specimens after type C1 RH (**A**) and TMMR (**B**). Author’s own material. A small part of lymphatic tissue of the paracervix is removed in figure A—the tissue below the ureter and above the deep uterine vein—together with the vein. UT—uterus; C—cervix; V—vagina; PuT—parauterine tissue; PrC—paracervix; USL—uterosacral ligament; RVL—rectovaginal ligament; VUL—vesicouterine ligament; VVL—vesicovaginal ligament; VM—vascular mesometrium; LM—ligamentous mesometrium.

**Table 1 cancers-15-05295-t001:** Differences regarding terminology and conception of anatomy between RH and TMMR.

Differences in Terminology and Conception of Anatomy	Radical Hysterectomy (C1/C2)	Total Mesometrial Resection
Avascular pelvic spaces	Exist	The concept of avascular spaces is not applicable for TMMR, despite some of the spaces developed during procedure (Latzko, Okabayashi, presacral)
Uterus ligaments	Exist	Only round and ovarian ligaments
Lateral parametrium		
Cranial part	Parauterine tissue	Vascular mesometrium Distal mesoureter Mesobladder
Caudal part	Paracervix	Mesobladder IHP Mesocolpos
Ventral parametrium		
Cranial part	Vesicouterine ligament	Mesobladder Vesicovaginal venous plexus
Caudal part	Vesicovaginal ligament	Mesobladder
Dorsal parametrium		
Cranial part	Sacrouterine ligament	Ligamentous mesometrium—upper part
Caudal part	Rectovaginal ligament	Ligamentous mesometrium—lower part
Lateral ligament of the rectum	Exist	Does not exist
Middle rectal artery	Exist	Does not exist
Mesoureter	In the same axis and part of the hypogastric nerve	Not part of the hypogastric nerve. The nerve is located between the mesoureter and ligamentous mesometria

**Table 2 cancers-15-05295-t002:** Autonomic nerve damage during parametrium and lymph node resection and functional outcomes.

Autonomic Nerve	Surgical Step	Functional Outcome
Lumbar splanchnic nerves	Lumbo-aortic lymph node dissection	Obstipation, anorectal dysfunction, sexual dysfunction (lubrication and orgasm disorders)
Superior hypogastric plexus	Lumbo-aortic lymph node dissection	Obstipation, anorectal dysfunction, sexual dysfunction (lubrication and orgasm disorders)
Hypogastric nerve	Rectouterine (uterosacral) ligament resection	Obstipation, flatus, anorectal dysfunction, lack of bladder-urine sensation
Pelvic splanchnic nerves	Paracervix resection	Lack of bladder-urine sensation and urinary retention, impaired voluntarily voiding, obstipation-anorectal dysfunction
Inferior hypogastric plexus	Paracervix and rectovaginal ligament resection	Lack of bladder-urine sensation and urinary retention, impaired voluntarily voiding, obstipation-anorectal dysfunction
Inferior hypogastric plexus rectal branches	Rectovaginal ligament resection	Obstipation, anorectal dysfunction
Inferior hypogastric plexus vesical fibers	Vesicovaginal ligament and paracolpium resection	Lack of bladder-urine sensation and urinary retention, impaired voluntarily voiding

**Table 3 cancers-15-05295-t003:** Differences in parametrial resection for type C1-C2 radical hysterectomy and TMMR.

Parametrium Tissue		Type C1	Type C2	TMMR
Lateral parametrium				
	Parauterine tissue	Excised from the origin at the level of the internal iliac artery	Excised from the origin at the level of the internal iliac artery	Excised from the origin at the level of the internal iliac artery
	Paracervix tissue	Partially removed, to the level of the deep uterine vein (vaginal vein) (IHP and PSN are preserved)	Totally removed, caudal to the deep uterine vein, to the level of levator ani (iliococcygeus) (IHP and PSN are sacrificed)	Paracervix is excised medial to mesoureter. Vaginal branches of IHP and vaginal veins are transected medial to the mesouterer
Dorsal parametrium				
	Rectouterine (uterosacral) ligament	Excised from the level of rectosacral angle (HN is preserved)	Excised from the level of rectosacral angle (HN is sacrificed)	Excised from the level of rectosacral angle (HN is preserved)
	Rectovaginal ligament	Partially excised—according to the resection plane of the vagina (IHP is preserved)	Excised from the level of pelvic parietal fascia (IHP is sacrificed)	Excised from the level of pelvic parietal fascia (IHP is preserved)
Ventral parametrium				
	Vesicouterine ligament (anterior leaf)	Excised medial to the distal ureter	Excised from the level of the bladder	Excised medial to the distal ureter
	Vesicovaginal ligament (vesicouterine ligament posterior leaf)	Excised above the IHP vesical fibers	Totally excised with the paracolpium to the level of pubococcygeus (IHP vesical fibers are sacrificed)	Not removed

## Data Availability

Authors declare that all the related data are available concerning the researchers.

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
