# Peer review of "Radical Hysterectomy or Total Mesometrial Resection—Two Anatomical Concepts for Surgical Treatment of Cancer of the Uterine Cervix"

_cancers, 2023, doi:10.3390/cancers15215295_

Round 1

Reviewer 1 Report

Comments and Suggestions for Authors

The efforts of the authors to clarify the surgical anatomy for the treatment of early cervix carcinoma are highly acknowledged. This reviewer does not comment on the “ligaments and spaces” concept as described in the manuscript, which is still the foundation of the worldwide treatment standard. Except: the length of the corresponding part of the text, its repetitive redundancy, the arbitrariness of the subperitoneal tissue regions indicated in the figures 1-6 (the unbordered irregular areas of fatty tissue in particular) are striking representations of the artificial nature of this confusing idea of the female subperitoneum.

As correctly stated by the authors tracing back the mature tissue derivatives to their stepwise development – ontogenetic anatomy – offers the key to understand (and to operate within) the complex subperitoneum in the female pelvis unbiased by dissection artifacts. In addition, ontogenetic anatomy allows to predict the potential extent of locoregional cancer spread which is necessary to plan the adequate surgical treatment and – also important – define its limitations for the individual malignant tumor.

Considering the ontogenetic anatomy of the female subperitoneum and its application in performing TMMR as described in the manuscript provokes the following statements:

1. The surgical goal of TMMR is the resection of the Müllerian compartment except its distal part preserving the subtotal vagina and the removal of all intercalated first-line lymph nodes. These lymph nodes that can be of microscopic size are located in the mesotissues derived from the anterior cloacal mesenchyme (termed vascular mesometrium and mesocolpos) and in the mesotissues derived from the distal splanchnic coelom (termed ligamentous mesometrium and mesocolpos). For locoregional control of cervix carcinoma TMMR is always combined with therapeutic lymph node dissection completely removing the basin nodes at risk according to an algorithm based on the ontogenetic tumor stage and intraoperative frozen section assessment. As adjuvant postoperative radiotherapy is strictly omitted with cancer field surgery locoregional recurrence analysis is the most accurate means to identify the lymph fatty tissues potentially infiltrated by cancer. All aspects of lymphatic drainage summarized by the authors at length are biased by their experimental set up and are of only limited informative strength.

Continuous local tumor propagation into and in-transit metastases within the mesotissues indicate that the potential cancer field exceeds the surgical treatment field and the surgical approach could be suboptimal. In these situations it is still not settled whether cancer field surgery or primary chemoradiotherapy achieve better treatment results.

2. The mesotissues (and the serous surfaces of the organs as well) as defined in ontogenetic anatomy are parts of mature compartments different from the corresponding organs.

3. The urogenital sinus is not a part of the cloacal membrane. The cloacal membrane is the precursor tissue of the surface of the phallic part of the urogenital sinus from which the vulva differentiates in the female. Both distal (sinus) vagina and urethra share a common mesenchyme mantle derived from the anterior cloacal mesenchyme. However, the sinus vagina is another subcompartment of the Müllerian compartment.

4. The so-called vesicouterine ligament comprises the bladder adventitia and the vesicovaginal venous plexus. It is part of another subcompartment of the anterior cloacal mesenchyme and does not contain intercalated lymph nodes for the drainage of the uterine cervix.

5. With TMMR the vascular mesometrium is completely resected up to its origin from the umbilical artery and bladder veins. The lateral transection plane begins above the ureter at the branching of the uterine artery but ends below the ureter level at the estuary of the uterine vein. However, the distal mesureter and the branches of the medially adjacent inferior hypogastric plexus are retained. The deep uterine or vaginal vein(s) and the Frankenhäuser plexus are transected medial to the distal mesureter to include the vascular mesocolpos with potential intercalated lymph nodes into the TMMR specimen. For cervix carcinomas confined to the Müllerian compartment, no tissue at risk for locoregional spread is retained.

Comments on the Quality of English Language

Minor editing of English language required

Author Response

Dear Reviewer,

We are deeply grateful for your comprehensive review. We incorporated the recommended changes. All incorporated changes are highlighted by using the Track and Changes in Word.

-…the length of the corresponding part of the text, its repetitive redundancy, the arbitrariness of the subperitoneal tissue regions indicated in the figures 1-6 (the unbordered irregular areas of fatty tissue in particular) are striking representations of the artificial nature of this confusing idea of the female subperitoneum.

Author’s Reply:

We agree with the reviewer. Actually, there was repetitive redundancy between paragraphs B and E, which was reduced (It could be easily seen by using Track and Changes). Paragraph D was also reduced.  We agree that figures 1-6 completely differentiate from the concept of the ontogenetic anatomy. However, the article aims to be objective when comparing the surgical anatomy of RH and TMMR. First part of the article demonstrates the concept of radical hysterectomy in the light of accepted classical anatomy. Therefore, we decided to perform a comprehensive review of the two surgical approaches without personal biases. Moreover, this is the reason why we decided to exclude as much as possible clinical data – the objectiveness will probably be lost.

- … In addition, ontogenetic anatomy allows to predict the potential extent of locoregional cancer spread which is necessary to plan the adequate surgical treatment and – also important – define its limitations for the individual malignant tumor.

Author’s Reply: We agree with the reviewer.

The next text was incorporated (red and italic):

Page: 16; Lines: 512-514

According to the theory, tumor growth is confined by a permissive compartment during a long time of its natural history and the compartment’s borders are tumor suppressive. It is  also a predictive model, which contributes for selecting the proper surgical procedure for every single patient according to the probability of compartments involvement.         

  1. The surgical goal of TMMR is the resection of the Müllerian compartment except its distal part preserving the subtotal vagina and the removal of all intercalated first-line lymph nodes. These lymph nodes that can be of microscopic size are located in the mesotissues derived from the anterior cloacal mesenchyme (termed vascular mesometrium and mesocolpos) and in the mesotissues derived from the distal splanchnic coelom (termed ligamentous mesometrium and mesocolpos). For locoregional control of cervix carcinoma TMMR is always combined with therapeutic lymph node dissection completely removing the basin nodes at risk according to an algorithm based on the ontogenetic tumor stage and intraoperative frozen section assessment. As adjuvant postoperative radiotherapy is strictly omitted with cancer field surgery locoregional recurrence analysis is the most accurate means to identify the lymph fatty tissues potentially infiltrated by cancer. All aspects of lymphatic drainage summarized by the authors at length are biased by their experimental set up and are of only limited informative strength.

Continuous local tumor propagation into and in-transit metastases within the mesotissues indicate that the potential cancer field exceeds the surgical treatment field and the surgical approach could be suboptimal. In these situations it is still not settled whether cancer field surgery or primary chemoradiotherapy achieve better treatment results.

Author’s Reply: We agree with the reviewer.

The next text was incorporated (red and italic):  

Page: 17;  Lines: 558 – 568

Two pairs of wings - vascular and ligamentous mesometria are noticed. These mesotissues (especially vascular mesometria) contain intercalated lymph nodes and should be removed during TMMR. Despite the proximity to the Mullerian compartment, mesometria originates from other compartments: vascular mesometria from anterior cloacal mesenchyme, and ligamentous mesometria from distal splanchnic coelom . In the context of TMMR, a few parts of urogenital sinus derivatives should be sharply dissected from the Mullerian and mesotissues compartments: dorsal bladder adventitia from the cervix, and the bladder mesentery from the vascular mesometrium. The vascular mesometrium contains the uterine artery, fibro-fatty and lymphatic tissue. 

Author’s Reply: We agree with the reviewer about the statement associated with therapeutic lymph node dissection. Although the pelvic lymph node dissection is not the topic of the present manuscript, it should be at least mentioned the difference between both procedures. Therapeutic role of lymph node dissection during TMMR is well known. Lymph node dissection during radical hysterectomy is a staging procedure.

The next text was inserted (red and italic):

Page: 28; Lines: 807-814

IT should be also stressed that during RH, pelvic and paraaortic lymphadenectomy is a staging procedure, that can be replaced by a sentinel lymph node biopsy. Moreover, the current guidelines for management of patients with cervical cancer recommends abandoning the hysterectomy in case of  lymph node metastases.  Contrarily, Höckel stated that  lymph node dissection during TMMR is therapeutic [47]. Intraoperative frozen section of lymph nodes is mandatory according to the TMMR algorithm and if metastases are found, the next line of lymph nodes must be removed. After lymphadenectomy, the TMMR is performed irrespectively of lymph node status.

All aspects of lymphatic drainage summarized by the authors at length are biased by their experimental set up and are of only limited informative strength.

Author’s Reply: We also agree with the statement about lymphatic drainage. Despite of artificial nature of ventral parametrium, we cannot ignore current, (although weak) evidence of its existence because this concept is accepted worldwide  and readers will definetly notice that we are not biased.

  1. The mesotissues (and the serous surfaces of the organs as well) as defined in ontogenetic anatomy are parts of mature compartments different from the corresponding organs.

Author’s Reply:

It is a really good statement. We incorporated it in the text.

The next text was incorporated (red and italic):

Page: 16; Lines: 519-520

In ontogenetic anatomy, the prefix “meso-“ defines a compartment derived from distinct primordial tissue. Mesotissues abut to corresponding organ, although, its origin differs from the organ.

  1. The urogenital sinus is not a part of the cloacal membrane. The cloacal membrane is the precursor tissue of the surface of the phallic part of the urogenital sinus from which the vulva differentiates in the female. Both distal (sinus) vagina and urethra share a common mesenchyme mantle derived from the anterior cloacal mesenchyme. However, the sinus vagina is another subcompartment of the Müllerian compartment.

Author’s Reply:

It is our mistake. We change this point.

The next text was incorporated (red and italic):

Page: 17; Line: 547

The urogenital sinus (part of the ventral cloaca) gives rise to the urinary bladder, urethra and distal 1/3 vagina. The intrapelvic part of this compartment has a well-demarcated border.

  1. The so-called vesicouterine ligament comprises the bladder adventitia and the vesicovaginal venous plexus. It is part of another subcompartment of the anterior cloacal mesenchyme and does not contain intercalated lymph nodes for the drainage of the uterine cervix.

Author’s Reply:

We completely agree with the statement of reviewer. Ventral parametrium is the most controversial part of traditional radical hysterectomy. These structures don’t have any clear resection margins and are obvious dissection artifacts. It was mentioned in the article. We also add information about vesicovaginal venous plexus in Table 1.

The next text was incorporated (red and italic):

Page: 18; Line : 609

It is a part of the bladder adventitia, and it belongs to the urogenital sinus compartment and does not contain intercalated lymph nodes related to the uterine cervix

  1. With TMMR the vascular mesometrium is completely resected up to its origin from the umbilical artery and bladder veins. The lateral transection plane begins above the ureter at the branching of the uterine artery but ends below the ureter level at the estuary of the uterine vein. However, the distal mesureter and the branches of the medially adjacent inferior hypogastric plexus are retained. The deep uterine or vaginal vein(s) and the Frankenhäuser plexus are transected medial to the distal mesureter to include the vascular mesocolpos with potential intercalated lymph nodes into the TMMR specimen. For cervix carcinomas confined to the Müllerian compartment, no tissue at risk for locoregional spread is retained.

Author’s reply:

This statement is well illustrated on Fig. 11. However, according to your recommendations we inserted the following text below figure 11 (red and italic) :

Vascular mesometrium is grasped by forceps and guided dorsally. Vaginal veins are ligated and transected medial to mesoureter. Distal mesoureter, hypogastric nerves and inferior hypogastric plexus are preserved.

We also changed Table 3 and added the following text (red and italic):

Paracervix is excised medial to mesoureter. Vaginal branches of IHP and vaginal veins are transected medial to the mesouterer.

Minor editing of English language required

Author’s Reply:

A native English speaker corrected errors in grammar, punctuation, word choice, and sentence construction to improve the flow of ideas expressed in the article.

We are grateful for your valuable time and effort in reviewing our manuscript.

Based on your useful and scientific comments, we believe our manuscript has been

improved to a higher level.

Reviewer 2 Report

Comments and Suggestions for Authors

The text is very didactic, with figures with excellent resolutions. I believe this is just a review in English, as there are parts that are a little confusing, especially in the section "Avascular Spaces nearby the Ventral Parametrium during Radical Hysterectomy".

The main question of the study is to compare classical and ontogenic anatomy in radical hysterectomy used to treat cervical cancer. It's not original work, but it's relevant. The authors carried out an extensive review using current data. In my opinion, they just need to review their English in the topics that I have highlighted previously. In my opinion, the figures have excellent resolution. Regarding the tables, I believe that table 1 deserves a better explanation, because it informs the difference between radical hysterectomy and total mesometrial resection. I was confused by the "+" signs.

Comments on the Quality of English Language

I believe this is just a review in English, especially in the section "Avascular Spaces nearby the Ventral Parametrium during Radical Hysterectomy".

Author Response

Dear Reviewer,

We incorporated the recommended changes. All incorporated changes are highlighted by using the Track and Changes in Word.

  1. The text is very didactic, with figures with excellent resolutions. I believe this is just a review in English, as there are parts that are a little confusing, especially in the section "Avascular Spaces nearby the Ventral Parametrium during Radical Hysterectomy".

Author’s Reply:

We agree with the reviewer. Therefore, a native English speaker corrected errors in grammar, punctuation, word choice, and sentence construction to improve the flow of ideas expressed in the article. Moreover, the section  "Avascular Spaces nearby the Ventral Parametrium during Radical Hysterectomy" was carefully revised in order to avoid confusion during reading.

  1. 2. The main question of the study is to compare classical and ontogenic anatomy in radical hysterectomy used to treat cervical cancer. It's not original work, but it's relevant. The authors carried out an extensive review using current data. In my opinion, they just need to review their English in the topics that I have highlighted previously. In my opinion, the figures have excellent resolution. Regarding the tables, I believe that table 1 deserves a better explanation, because it informs the difference between radical hysterectomy and total mesometrial resection. I was confused by the "+" signs.

Author’s Reply:

We agree with the reviewer. We made changes of the table. Moreover signs such as “+” and “-“were replaced by the words – “exist” or “do not exist” according to the conceptions of classic anatomy or ontogenetic anatomy.

  1. I believe this is just a review in English, especially in the section "Avascular Spaces nearby the Ventral Parametrium during Radical Hysterectomy".

Author’s Reply:

We agree with the reviewer. Therefore, a native English speaker corrected errors in grammar, punctuation, word choice, and sentence construction to improve the flow of ideas expressed in the article.

We are grateful for your valuable time and effort in reviewing our manuscript.

Based on your useful and scientific comments, we believe our manuscript has been

improved to a higher level.

Reviewer 3 Report

Comments and Suggestions for Authors

I read with great interest the manuscript, which falls within the aim of this Journal and offers a high-quality overview of the topic. The abstract is complete and asaustive. The figures are detailed and allow an immediate understanding.
Although the manuscript can be considered already of high quality, I would suggest taking into account the following minor recommendations:

- I recommend adding further details to discuss the ongoing debate about the use/avoidance of minimally invasive surgery for cervical cancer, considering the available pieces of evidence and the histotype of the disease (authors may refer to: PMID: 36293758).

The data presented makes the manuscript interesting, the authors have performed a deep and methodologically rigorous review, and offered clear and balanced conclusions to the readers, addressing future research priorities. For all these reasons, I recommend the publication of the article, pending a few minor revisions.

Comments on the Quality of English Language

Minor editing of the English language is required to make the work clearer and more readable.

Author Response

Dear Reviewer,

We incorporated the recommended changes. All incorporated changes are highlighted by using the Track and Changes in Word.

I read with great interest the manuscript, which falls within the aim of this Journal and offers a high-quality overview of the topic. The abstract is complete and asaustive. The figures are detailed and allow an immediate understanding.

Although the manuscript can be considered already of high quality, I would suggest taking into account the following minor recommendations:

  1. 1. I recommend adding further details to discuss the ongoing debate about the use/avoidance of minimally invasive surgery for cervical cancer, considering the available pieces of evidence and the histotype of the disease (authors may refer to: PMID: 36293758).

Author’s Reply:

We agree with the reviewer. The role of minimally invasive surgery should be stated, as there are many controversies after the LACC trial. Although clinical data are not the topic of the present manuscript, the debate about minimally invasive versus open surgery ( both in RH and TMMR) should be at least mentioned. This debate will also explain to the readers why all of the surgical figures in the manuscript are done by open surgery. The recommended reference (PMID: 36293758) was incorporated.  

We stated the histotypes of the disease during the LACC trial, but further explanation will add to this chapter too much clinical data, which as it was mentioned (in the abstract and introduction) is not the topic of the manuscript.

The next text was incorporated:

Page: 23; Line: 714-725;

RH and TMMR show worse patients' outcomes (higher recurrence rate, inferior overall survival)  when they are performed by minimally invasive approach compared to open abdominal surgery. Höckel always performs TMMR by open surgery. Moreover, one study reported that minimally invasive TMMR (laparoscopic and robotic) is associated with a higher recurrence rate compared to open surgery. The Laparoscopic Approach to Cervical Cancer Trial (LACC trial) showed that minimally invasive RH was associated with inferior disease-free survival and overall survival compared to open abdominal RH for patients with cancer of the uterine cervix (stages IA1 with lymphovascular invasion, IA2, IB1 and histologic subtypes of squamous-cell carcinoma, adenocarcinoma, or adenosquamous carcinoma). However, the debate about the role of minimally invasive RH for the treatment of cervical cancer is still ongoing. Nevertheless, open RH is the standard approach in most centers.

The next references were incorporated

Ramirez, P. T., Frumovitz, M., Pareja, R., Lopez, A., Vieira, M., Ribeiro, R., Buda, A., Yan, X., Shuzhong, Y., Chetty, N., Isla, D., Tamura, M., Zhu, T., Robledo, K. P., Gebski, V., Asher, R., Behan, V., Nicklin, J. L., Coleman, R. L., & Obermair, A. (2018). Minimally Invasive versus Abdominal Radical Hysterectomy for Cervical Cancer. The New England journal of medicine, 379(20), 1895–1904. https://doi.org/10.1056/NEJMoa1806395

van de Lande, J., von Mensdorff-Pouilly, S., Lettinga, R. G., Piek, J. M., & Verheijen, R. H. (2012). Open versus laparoscopic pelvic lymph node dissection in early stage cervical cancer: no difference in surgical or disease outcome. International journal of gynecological cancer : official journal of the International Gynecological Cancer Society, 22(1), 107–114. https://doi.org/10.1097/IGC.0b013e31822c273d

Pecorino, B., D'Agate, M. G., Scibilia, G., Scollo, P., Giannini, A., Di Donna, M. C., Chiantera, V., & Laganà, A. S. (2022). Evaluation of Surgical Outcomes of Abdominal Radical Hysterectomy and Total Laparoscopic Radical Hysterectomy for Cervical Cancer: A Retrospective Analysis of Data Collected before the LACC Trial. International journal of environmental research and public health, 19(20), 13176. https://doi.org/10.3390/ijerph192013176

Cibula D, Raspollini MR, Planchamp F, et alESGO/ESTRO/ESP Guidelines for the management of patients with cervical cancer – Update 2023*International Journal of Gynecologic Cancer 2023;33:649-666.

The data presented makes the manuscript interesting, the authors have performed a deep and methodologically rigorous review, and offered clear and balanced conclusions to the readers, addressing future research priorities. For all these reasons, I recommend the publication of the article, pending a few minor revisions.

  1. Minor editing of the English language is required to make the work clearer and more readable.

Author’s Reply:

A native English speaker has revised the article in order to improve English gramma.

We are grateful for your valuable time and effort in reviewing our manuscript.

Based on your useful and scientific comments, we believe our manuscript has been

improved to a higher level.

Reviewer 4 Report

Comments and Suggestions for Authors

Dear Editors/Authors

Thank you for giving me the opportunity to review this review article. Below this is my comment

·       This is a review article that focuses on the comparison between the anatomical and surgical basics of radical hysterectomy (type C1/C2) and total mesometrial resection (TMR). The authors do an excellent work to review previous publications about this issue. However, I would like to suggest the authors mention the origin of all figures in this article.

·       The review is clear, comprehensive, and of relevance to the radical hysterectomy field. The readers should understand the relation between these 2 terminologies (radical hysterectomy and TMR classification). To the best of my knowledge, there has been no review published like this before.

·       The cited references were mostly recent publications and relevant to the article and did not include an excessive number of self-citations

·       The statements and conclusions drawn are coherent and supported by the listed citations

·       All the figures and tables were appropriate and clearly understood the labels. However, the authors should provide the origin of these figures.

 Best Regards

Dr. Prapaporn Suprasert

Author Response

Dear Reviewer,

We incorporated the recommended changes. All incorporated changes are highlighted by using the Track and Changes in Word.

Thank you for giving me the opportunity to review this review article. Below this is my comment

  1. This is a review article that focuses on the comparison between the anatomical and surgical basics of radical hysterectomy (type C1/C2) and total mesometrial resection (TMR). The authors do an excellent work to review previous publications about this issue. However, I would like to suggest the authors mention the origin of all figures in this article.

Author’s Reply:

We agree with the reviewer. However, below every surgical figure, it is written either “author’s own material” or “surgical dissection by author IS”. The authors did all of the surgical or cadaveric figures. The figures with drawings are also author’s material. Only one figure (figure 7 – Carnegie stage female embryo) in the manuscript is used from another source. We obtained permission and cited the source. It is written below the figure (adopted with permission from reference 37].

 To avoid confusion among readers the next text was incorporated in the section “ethical statement”:

“The intraoperative or cadaveric figures are from surgical cases or cadaveric dissections of the represented authors”

  • The review is clear, comprehensive, and of relevance to the radical hysterectomy field. The readers should understand the relation between these 2 terminologies (radical hysterectomy and TMR classification). To the best of my knowledge, there has been no review published like this before.
  • The cited references were mostly recent publications and relevant to the article and did not include an excessive number of self-citations
  • The statements and conclusions drawn are coherent and supported by the listed citations
  • 2. All the figures and tables were appropriate and clearly understood the labels. However, the authors should provide the origin of these figures.

Author’s Reply:

It was explained above.

We are grateful for your valuable time and effort in reviewing our manuscript.

Based on your useful and scientific comments, we believe our manuscript has been

improved to a higher level.

Round 2

Reviewer 1 Report

Comments and Suggestions for Authors

The authors respected this reviewer's recommendations